# Document Classification using Reference Information

## Abstract

Document classification is a common problem when organizing free-text data, but supervised classification algorithms often require many labeled examples and tedious manual annotation by humans labelers. This work propose an innovative methodology called Document Classification using Reference Information (DCRI), which classifies documents with little human intervention by leveraging the existence of reference information, documents from external sources related to the label classes of interest. For example, when classifying news articles into topics, Wikipedia articles can serve as such an external source. DCRI uses reference information to generate weak initial labels for an unlabeled corpus, then iteratively augments them into stronger labels using both supervised machine learning algorithms and limited human labeling capacity, if available. DCRI is evaluated on one dataset from a major pharmaceutical manufacturing company and two public datasets for news topic classification. When no human labeling capacity is available, DCRI achieves an accuracy between 84% to 96% on these three datasets. When some manual labeling capacity is available, DCRI helps prioritizing labeling documents with high uncertainty. To shed light on the value of reference information, this paper also develops a generative mathematical model in which reference information provides a noisy estimate of the latent distribution that generates documents. An extensive numerical study is performed using synthetic data to analyze when and why reference information is most valuable. Finally, for a special case of the model with two classes, a theoretical result is established to show the value of the iterative nature of the DCRI approach.

## 1 Introduction

Over the past few decades, digitization has led to an abundance of data in almost every industry. Even as the amount of available structured data increases rapidly, unstructured free-text documents are often an untapped important source of information, especially in environments where text is generated manually by human experts. Unlike numerical and categorical features in tabular form, a corpus of documents is difficult to analyze because it requires extracting meaningful insights and patterns from free-text data. The ability to classify documents into a more structured schema is essential for faster information retrieval and trend analysis. A common approach to impose structure on such free-text documents is to apply unsupervised clustering algorithms, which detect underlying similarities without any labeled target. Although such an approach involves minimal manual intervention, the resulting clusters are often difficult to interpret and are often unstable. However, in many real-world applications, subject matter experts (SME) often have a schema of meaningful labels in mind that would be most useful to them. An unsupervised algorithm is unlikely to produce clusters that exactly match the desired label schema. To match the desired schema, it is often required to apply a supervised text classification algorithm trained on examples of documents from the desired label schema. Unfortunately, supervised algorithms rely heavily on labeling training examples, which typically require substantial manual effort by SMEs.

Motivated by this challenge, this paper aims to develop an algorithmic methodology to classify a large corpus of documents into the desired taxonomy of classes *efficiently and accurately* in the absence of existing training data. In particular, this paper proposes a novel methodology called Document Classification using Reference Information (DCRI), which can produce high-quality labels with minimal human intervention. A key aspect

| Unlabeled Corpus Documents | Desired Classes | Example Classes | Reference Information Documents |
|---|---|---|---|
| Deviation reports in the pharmaceutical manufacturing line | Process steps | Filter integrity testing, sampling | Standard operating procedure documents |
| News articles | Topic | Politics, business, sports | Wikipedia articles |
| Legal documents | Type | Affidavit, contract, patent | Writing guidelines (e.g., manuals and templates) |
| Research papers | Subject | Algorithms, data science, optimization | Textbooks |

Table 1: Examples of Corpus Documents and Reference Information

of DCRI is leveraging external documents called *reference information documents* to quickly produce weak labels for the unlabeled corpus of documents through a nearest neighbors based classifier. Broadly speaking, reference information refers to documents outside the unlabeled corpus that are typically related to the labels of interest. For example, to classify news articles into topics such as business, politics, science...etc, Wikipedia articles titled business, politics, science...etc. They are sufficiently similar to their news article counterparts and can provide a weak initial label for each news article by using a simple nearest neighbors classifier. That is, each document in the corpus is assigned a label according to which Wikipedia article it is "closest" to (i.e., most similar under some metric). Other examples of document and reference information pairs are shown in Table 1, for example classifying legal documents by type (e.g., contract, affidavit, patent) using law school writing manuals, or research papers by topic using academic textbooks. One major advantage of using reference information is that minimal manual work is needed as reference information tend to be much longer in length than each unlabeled document, and thus only a relatively small number of sources of reference information per class is sufficient to generate weak labels that are much better than random guessing.

Since *reference information documents* could be very different from the unlabeled *corpus documents*, the labels generated by a nearest neighbors classifier will typically be far from perfect, although much better than random guessing. The DCRI methodology continues to improve the initial labels using iterations of supervised classification, starting with the initial label as the dependent variable. In each iteration, a supervised model is trained with the labels from the previous iteration as the dependent variables, and used to predict new labels. This process is repeated for a specified number of iteration or until the label do not change anymore. Such an iterative approach can fine-tune the decision boundary between classes and is shown to greatly improve the accuracy of the predicted labels. Finally, if some labeling capacity from SME's is available, DCRI can strategically incorporate these labels to further improve the accuracy of the labels.

This work is motivated by a collaboration with a major pharmaceutical manufacturing company. We describe the setting, need for labeled documents and the availability of reference information for this particular application.

## 1.1 Motivating Use Case: Pharmaceutical Manufacturing Line

The manufacturing of pharmaceutical drugs, especially biologics (i.e., drugs grown from living cell cultures), involves intricate processes that take place using many complex process steps and equipment. Figure 1 shows a schematic depiction of the process steps and equipment (numbered 1 through 7) used in a typical manufacturing of biologic drugs, including inoculation of cells, growth in bioreactors, filtration and final product assembly. To ensure quality and safety of the drug, the manufacturing process is highly regulated by the US Food and Drug Administration (FDA). One requirement imposed by the FDA is to thoroughly document any deviation from these protocols and perform the appropriate investigations to identify the

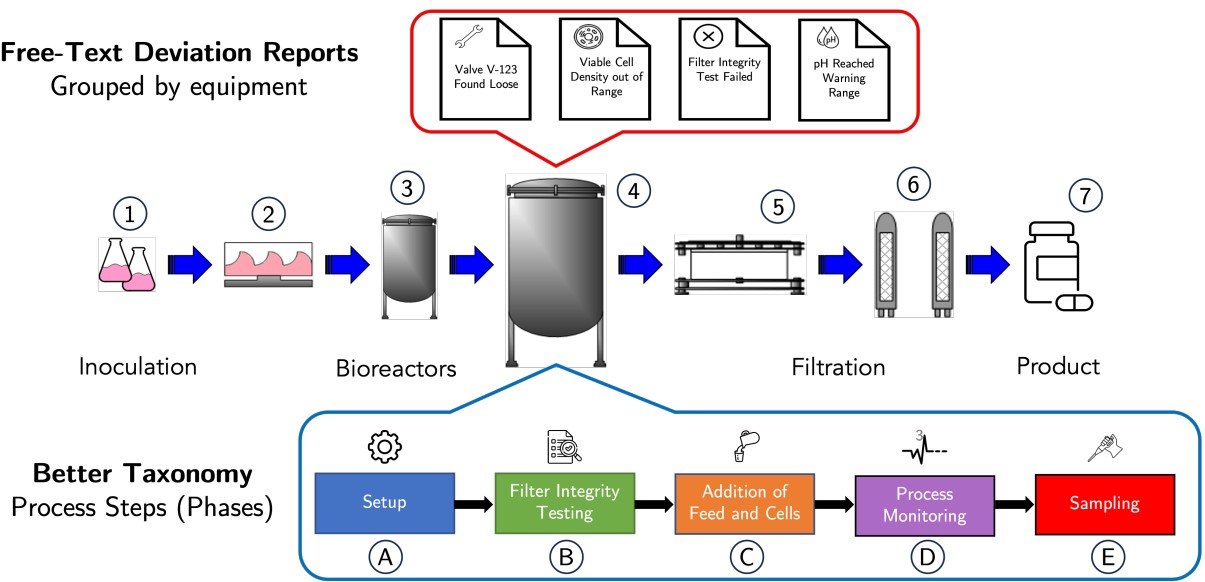

Figure 1: Equipment $1-7$ and the Process Steps for Equipment 4

root cause and mitigate future deviations (U.S. Department of Health and Human Services (b;a)). As a result, a pharmaceutical manufacturing company typically holds thousands of deviation reports for each manufacturing line, with most of these being minor deviations that do not likely affect the product's quality.

Besides satisfying regulatory requirements, this corpus of deviation reports is potentially useful in two ways, (1) to expedite investigation of future deviations and (2) to detect trends and repeated patterns. To facilitate these two goals, the deviation reports have a very coarse classification that associates each deviation report a respective piece of equipment (numbered 1 through 7) that the deviation occurred on. For example, the top of Figure 1 shows some sample deviations associated with equipment number 4, the main bioreactor. This organization allows for faster information retrieval as an investigator can quickly pull up all deviations that occurred on a particular bioreactor. Furthermore, an analyst can easily see that deviations in a particular bioreactor are trending higher, implying a potential degradation in the equipment's quality.

Experts developed a more fined-grained taxonomy that associates each deviation report with the respective *process step* it occurred during. The bottom of Figure 1 shows the 5 process steps (labeled A through E) associated with the main bioreactor (equipment 4). These process steps include setup (A), testing of crucial components (B), addition of cells (C), monitoring their health (D) and sampling cells to test for desired cell density (E). Classifying deviation reports to their respective process step is more useful because deviations associated with the same process step are significantly more likely to be related. In addition, there are common process steps across similar equipment, for example all bioreactors having a process monitoring phase.

Unfortunately, since this was a new taxonomy, there were no labeled examples under this taxonomy to use to train a model to automate the labeling process. Instead, the approach was to leverage the fact that the regulatory requirements impose very detail documentation of the nature of each process step in the form of Standard Operating Procedures (SOP)'s documents. An SOP is a detailed, step-by-step guide that describes how a human operator should perform or monitor a particular process step, including expected outcomes and common troubleshooting directions. SOP's are extremely common in the pharmaceutical manufacturing industry to ensure consistency and quality. Thus, such documents were a natural choice as reference information documents.

## 1.2 Contributions

Next, the main contributions of this paper are outlined.

**Novel Algorithmic Framework (Section 3).** The paper describes DCRI as a new conceptual algorithmic framework to classify unlabeled corpus of documents, leveraging side information in the form of reference documents. If some expert labeling capacity is available, the DCRI framework can incorporate those expert labels in an efficient manner. The framework is robust in that it is modular and can incorporate a range of algorithmic implementations. To the best of our knowledge, this work is the first to propose the use of reference information to generate weak labels for an entire dataset. The iterative procedure of improving the labels is inspired by work in Semi-Supervised Naive Bayes (Sristy & Somayajulu (2012); Zhao et al. (2016)), while the manner by which expert labels are incorporated is inspired by the literature on active learning Monarch (2021). However, this paper differs in that it uses weak labels generated from an external source while existing work uses strong labels generated by SMEs.

**Results on Real-World Data (Section 4).** We study a particular algorithmic implementation of DCRI that leverages a bag of words embedding and the Multinomial Naive Bayes classifier. We apply algorithm, called *DCRI with MNB*, to 3 real-world datasets, one in pharmaceutical manufacturing and two in news topic classification and compare it with unsupervised, supervised and semi-supervised approaches. With no labeled training examples, DCRI with MNB achieves an final accuracy of 83%, 89% and 96% for each of the three datasets, while an unsupervised approach that does not utilize reference information is only able to achieve an accuracy 52%, 55% and 78% respectively. Furthermore, DCRI with MNB's final accuracy is within 8% of what a supervised Multinomial Naive Bayes classifier can achieve if training data were available, implying that DCRI's accuracy is already quite close to the best accuracy any automated algorithm can hope to obtain. DCRI with MNB without any labeled training examples is able to match the performance of a semi-supervised approach with between 75 and 330 labeled examples. When some expert labeling capacity is available, DCRI with MNB can prioritize which documents require a manual review and outperforms existing semi-supervised approaches that randomly select which documents to label. By manually reviewing 10% of each corpus, prioritized by DCRI with MNB's confidence, the accuracies improve to 88.5%, 92.5% and 98.3%, respectively. This is significantly higher than choosing 10% of the corpus uniformly at random, which results in accuracies of only 84.8%, 90.1% and 95.9%, respectively.

**Generative Model and Numerical Study (Sections 5).** To further explain why DCRI performs well, we develop a generative mathematical model called the *Noisy Vocabulary Model*. This model is inspired by data from real-world settings. In particular, documents from each class are drawn from some latent distributions, while the reference documents information provides a *noisy*, distorted estimate of those distributions. Corpus documents and reference information documents are likely written in potentially very different contexts (e.g., news articles vs. Wikipedia). Therefore, the distributions they induce over the vocabulary space are different but with some overlap. This overlap is large enough such that initial weak labels are much better than random guessing, but the iterative label updating process is crucial to improve their quality.

Using synthetic data generated from our mathematical model, we perform an extensive numerical study on the value of reference information in comparison to unsupervised and supervised benchmark approaches. We study the performance of DCRI while varying the quality of the reference information (i.e., amount of noise inserted), number of documents and the class balance of the labels. Our findings confer and extend the insights from the real-world dataset.

**Theorem: Iteration Improves Accuracy (Section 6).** For a special case of our generative model with $K = 2$ classes, we prove that treating the initial labels as ground truth and training a supervised classifier results in an improvement in label accuracy (Theorem 6.1). Broadly speaking, the iterated labels improves upon the initial labels by correcting mistakes (i.e., the noise) from the reference information. We prove that the decision boundary between label class 1 and 2 is updated in a way that corrects mistakes and thus improves accuracy.

**Managerial Insight.** Our work implies that regardless of the application or the amount of human labeling capacity available, it is always worth considering whether reference information can be easily curated. There is significant value in being able to quickly produce a weak label for all documents, which can then be augmented using a combination of the iterative procedure as well as strategically using expert labels.

The rest of our paper is organized as follows. Section 2 presents related work regarding supervised, unsupervised and semi-supervised classification approaches. Section 3 describes the DCRI methodology while section 4

provides the results from that methodology on real-world datasets. Motivated by these results, Sections 5 presents a data generating model and the results of DCRI on synthetic data. Finally, Section 6 presents a theoretical result for a special case of the data generating model.

## 2   Related Work

Document classification is a subject with a long history of literature. Broadly speaking, document classification can be broken down into three settings: supervised, unsupervised and semi-supervised.

**Supervised.** Algorithms for supervised text classification are trained on a corpus of labeled documents typically produced by human experts. While supervised algorithms can be very accurate at document classification tasks such as sentiment analysis, spam detection and news topic classification, a major downside of such approaches is that they typically require a large amount of training data, which may not be accessible in all applications.

The most common algorithms for text classification are Naive Bayes (Lewis (1998)), Support Vector Machines (Vapnik & Chervonenkis (2015)) and K-Nearest Neighbors (Fix & Hodges Jr (1951)). Naive Bayes makes the assumptions that features (i.e. words in the text) are are drawn independently from each other and uses this assumption to estimate generative distributions for data from each class. Naive Bayes is commonly used in text classification due to its tractability in high dimensions (see e.g. Kibriya et al. (2005); Abbas et al. (2019); Wang et al. (2015); Frank & Bouckaert (2006)). Support Vector Machines (SVM) is a classification algorithm that separates data in the training set using a hyperplane that maximizes the margin, i.e. distance between the hyperplane and nearest data point from each class. SVM's have been successful at text classification as they handle high-dimensional and sparse data well, see e.g. Fatima et al. (2017); Mohammad et al. (2016); Hao et al. (2009); Rennie & Rifkin (2001). Finally, K-Nearest Neighbors is a simple classification algorithm which assigns labels to unseen data by taking the majority vote out of the K closest data points in the training set using some pre-specified distance metric. KNN is often used in text classification due to its simplicity and interpretability, see e.g. Khamar (2013); Han et al. (2001); Jiang et al. (2012); Kurada & Pavan (2013); Bijalwan et al. (2014); Kwon & Lee (2003). See also Kadhim (2019) for a survey on supervised approaches for text classification.

**Unsupervised.** Unsupervised approaches to document classification discover patterns in documents without any external guidance from subject matter experts. While unsupervised approaches obviate the need for large numbers of training examples, the clusters that are produced by such algorithms do require human interpretation. Such clusters may not always be interpretable to a human as there is little control over what exact similarities between documents are identified.

Some examples of algorithms for unsupervised learning include Latent Dirichlet Allocation (Blei et al. (2003)), Latent Semantic Analysis (Landauer et al. (1998)) and K-Means (Lloyd (1982)). Latent Dirichlet Allocation (LDA) discovers topics in a collection of documents by imposing a Bayesian generative model for each document, controlled by a set of latent topics. Similarly, Latent Semantic Analysis (LSA) discovers patterns in the documents by performing a matrix-factorization like procedure on the document-word matrix and uncovering a low-dimensional latent space of topics. Both LDA and LSA have been widely successful in discovering topics from unlabeled data (see e.g. Griffiths & Steyvers (2004); Foong & Ismail (2020); Yau et al. (2014); Sun (2014); Xie & Xing (2013); Dumais (2004); Wei et al. (2008); Song & Park (2007)). Finally, the widely popular K-Means algorithm for general clustering can be applied to text classification as well and has been shown to be successful (see e.g. Singh et al. (2011); Ferdous et al. (2009); Chouhan & Purohit (2018)).

**Semi-Supervised.** While supervised and unsupervised learning are two extreme ends in terms of the amount of human labor needed, semi-supervised learning represents a middle-ground where some manual work by humans is needed, but not to label a large corpus of documents. See van Engelen & Hoos (2020) or Zhu (2005) for a survey of semi-supervised learning. A common methodology in semi-supervised settings is to manually label a small subset of the data, build a supervised classifier using that subset, predicting labels for the unlabeled data and finally training another classifier on the all the labels, including the newly produced labels. In this setting, it has been shown that the unlabeled documents can provide substantial value over training a classifier on the originally labeled examples alone because the unlabeled examples can

carry information that is not present in the labeled data. One such semi-supervised approach that bears some resemblance to our DCRI methodology is semi-supervised Naive Bayes Nigam et al. (2006); Sristy & Somayajulu (2012); Zhao et al. (2016). Using a small number of labeled documents, semi-supervised Naive Bayes first trains a supervised classifier on these labeled documents and uses it to generate a weak label for the unlabeled documents. Then, treating these unlabeled documents as true labels, we iteratively train a supervised Naive Bayes classifier and re-predict the labels of the originally unlabeled documents. The DCRI methodology can be thought of as an extension of semi-supervised Naive Bayes in which instead of obtaining weak labels from a small set of labeled examples, we obtain weak labels from an external source, the reference information. Crucially, DCRI does not require an expert to label any documents at all. Section 4.3 compares the DCRI methodology to semi-supervised Naive Bayes and shows that reference information can substitute for hundreds of labeled documents.

Besides manually labeling documents, subject matter experts can be involved in other ways. One line of work tries to guide an unsupervised learning algorithm to a desired set of classes by having subject matter experts manually identify constraints on pairs of documents that must be in the same cluster and pairs that must not. See e.g. Ji & Xu (2006); Vilhagra et al. (2020); Huang et al. (2007). Another line of work involves having subject matter experts label *features* (i.e. words) instead of documents themselves (see e.g. Haj-Yahia et al. (2019); Kadar & Iria (2011); Nourashrafeddin et al. (2013)). Finally, a major area of research in semi-supervised learning is the field of active learning. Active learning is a machine learning paradigm that aims to reduce the number of examples needed to train a supervised algorithm by selectively annotating examples by querying a subject matter expert in an interactive fashion. It is well known that by querying the label of samples which one is least confident amount (i.e. on the margin), fewer examples are needed to reach the same out-of-sample performance. Some examples of work on active learning for text classification are Tong & Koller (2001); Goudjil et al. (2018); Li et al. (2012); Figueroa et al. (2012). See also Monarch (2021) for an entire textbook on active learning. When some expert labeling capacity is available, DCRI applies a similar principal as active learning, choosing to label documents are the least confident. However, this confidence score is determined from reference information rather than from a small set of expert labeled documents. DCRI does not require any expert labeled documents at all to obtain such a confidence score. To the best of our knowledge, literature in active learning has not studied the case when the confidence scores is given from a noisy external source.

## 3 The DCRI Framework

The DCRI methodology broadly refers to a modular algorithmic framework for generating labels for a large corpus of documents using a minimal amount of manual work from SMEs. A high-level overview of this framework is provided below in Section 3.1, while Section 3.2 describes the specific implementation that is used to generate the experimental results shown in the rest of the paper.

### 3.1 General Framework

**Input and Output.** The input to DCRI consists of: (1) a set of unlabeled *corpus documents* and (2) a labeling scheme. The output of DCRI is a predicted label for each corpus document under the desired labeling scheme. DCRI seeks to produce accurate labels, ones that match what an SME would assign if all corpus documents were manually labeled.

**Expert Involvement.** While DCRI is mostly an automated algorithm, human SME's are involved in two ways. First, SMEs are asked to curate the reference information document for each label class. Recall that reference information refers to documents related to the desired label classes, but are not corpus documents themselves. This typically involves a modest amount of laborious work as the number of label classes is small and SME's are familiar potential sources of reference information. Second, DCRI can naturally incorporate any amount of available labeling capacity in which SMEs directly label corpus documents. As shown later in the empirical results, even a small amount of optimally allocated labeling capacity can greatly improve the output label accuracy. We now dive into an overview of the DCRI framework. Note that these phases are intentionally described very broadly here, while Section 3.2 describes a specific implementation for each phase.

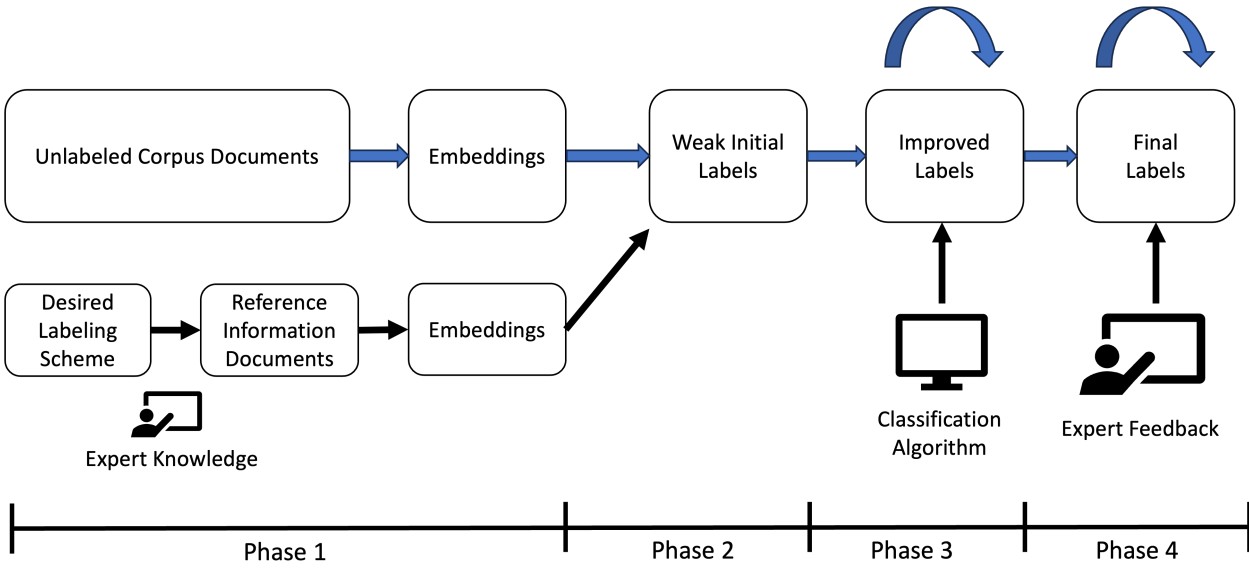

Figure 2: Phases of the DCRI framework

**Phases of the Algorithm.** DCRI can be broken down into a preprocessing phase followed by three algorithmic phases. Figure 2 provides a graphical overview of these four phases, which are described below.

- **Phase 1: Gather Reference Information and Vectorization.** For each label class, ask an SME to find relevant reference information documents. Without loss of generality, assume that each label class has a single reference information document, which can be the concatenation of multiple relevant documents. Then, embed all corpus documents and reference information documents as numerical vectors, a crucial phase in any natural language processing task.

- **Phase 2: Weak Initial Label.** Leverage the embeddings of the reference information documents to generate a weak initial label for each corpus document through nearest neighbors in the embedding space. This is done by assigning each document a label based on which reference information document is the "closest" to that document under some measure of similarity.

- **Phase 3: Improve Label Iteratively** Train a supervised classifier on the corpus documents' embeddings using the initial labels as the dependent variable. Then, take this classifier and predict new labels on each of the corpus documents. Repeat this process for a fixed number of iterations.

- **Phase 4: Incorporate Expert Feedback** Ask an SME to label a subset of the corpus documents, correcting some of the improved labels from Phase 3, then iteratively the labels again in a similar fashion as Phase 3.

Phase 2 generates weak labels by leveraging similarities between reference information documents and corpus documents, despite being from different sources (e.g., Wikipedia articles and news articles). While the weak labels from Phase 2 only depend on the reference information documents, Phase 3 leverages the structure of the unlabeled corpus documents itself to improve labels. Finally in Phase 4, by strategically choosing which labels from Phase 3 require manual inspection, DCRI can strategically utilize limited SME labor to achieve significant gains in accuracy from the Phase 3 labels.

## 3.2 DCRI with Multinomial Naive Bayes

This section describes the specific implementation of the DCRI framework studied in this paper, which leverages a bag of words (BoW) embedding for vectorization and a Naive Bayes Classifier (MNB). We refer

| Symbol | Meaning |
|--------|---------|
| $N \in \mathbb{N}$ | Number of unlabeled corpus documents |
| $K \in \mathbb{N}$ | Number of classes |
| $V \in \mathbb{N}$ | Size of vocabulary space |
| $X_{nv}$ | Count of word $v$ in corpus document $n$ |
| $L_n$ | Length of corpus document $n$, i.e. $\sum_{v \in [V]} X_{nv}$ |
| $a \in [N]$ | Smoothing factor |
| $r_{kv}$ | Proportion of word $v$ in reference information document $k$ (smoothed to always be non-zero) |
| $y_n^{(0)} \in [K]$ | Initial label of corpus document $n$ from reference information only. |
| $T \in \mathbb{N}$ | Number of iterations before querying an expert. |
| $y_n^{(t)} \in [K]$ | Iterated label of corpus document $n$ after $t$ iterations. |
| $\theta_k^{(t)} \in \Delta^V$ | Estimated distribution of class $k$ corpus documents at iteration $t$ |
| $y_n^{(T)} \in [K]$ | Iterated label before querying expert. |
| $p_n^{(T)} \in [0,1]$ | Estimated probability associated with $y_n^{(T)}$ |
| $Q \in \mathbb{N}$ | Number of expert labels we are allowed to query |
| $T' \in \mathbb{N}$ | Total number of iterations ($T' > T$) |
| $y_n \in [K]$ | True label (unobserved), but provided by an SME if queried. |
| $y_n^{(T')} \in [K]$ | Final label output of DCRI. |

Figure 3: Notation Overview

to this implementation as *DCRI with MNB* for short. Specifically, DCRI with MNB uses the following implementation for each of the 4 phases.

- **Phase 1 (Section 3.2.1):** Bag of words vectorization.

- **Phase 2 (Section 3.2.2):** Use KL Divergence between empirical vocabulary distributions as the measure of closeness.

- **Phase 3 (Section 3.2.3):** Iteratively apply a Multinomial Naive Bayes classifier.

- **Phase 4 (Section 3.2.4):** Prioritize expert labels based on predicted probability (confidence) from MNB.

All notation defined in the rest of this subsection is summarized in Figure 3. Pseudocode for Phase 2, 3 and 4 of the DCRI with MNB methodology is provided in Algorithm 1 and all line numbers mentioned in this section refer to this code block.

### 3.2.1 Phase 1: Vectorization with Bag of Words.

After SME's gather reference information documents, DCRI with MNB vectorizes both corpus and reference information documents using a *bag of words* embedding, a simple yet commonly used vectorization approach that represents a document as a count over tokens. This process involves basic text cleaning of the corpus, defining a large set of tokens (i.e., words or part of words), filtering the set of tokens down to a manageable size, then constructing the token counts of each document. Appendix B.1 provides more details of this process. We colloquially refer to tokens as *words* and the set of filtered tokens to be the *vocabulary space*.

---

**Algorithm 1** DCRI with Multinomial Naive Bayes (MNB)

---

1: **Input:** $N, K, V, Q, X_{nv}, X_{nv}, R_{kv}, r_{kv}$

2: **Output:** $\left\{ \hat{y}_n^{(T')} \right\}_{n \in [N]}$

3: **Parameters:** $T, T'$

4: $y_n^{(0)} \leftarrow \arg\max_{k \in [K]} \sum_{v \in [V]} X_{nv} \log(r_{kv})$

5: **for** $t = 1, 2, \ldots, T$ **do**

6: $\quad \theta_{kv}^{(t)} = \dfrac{\sum_{n=1}^{N} \mathbb{I}\{y_n^{(t-1)} = k\} \cdot X_{nv}}{\sum_{n=1}^{N} \mathbb{I}\{y_n^{(t-1)} = k\} \cdot L_n}$

7: $\quad y_n^{(t)} \leftarrow \arg\max_{k \in [K]} \sum_{v \in [V]} X_{nv} \cdot \log(\theta_{kv}^{(t)})$

8: **end for**

9: $p_n^{(T)} \leftarrow \dfrac{\prod_{v=1}^{V} (\theta_{k_n^* v})^{X_{nv}}}{\sum_{k=1}^{K} \prod_{v=1}^{V} (\theta_{kv})^{X_{nv}}}, \ k_n^* = y_n^{(T)}$

10: $\mathcal{Q} \subset [N] \leftarrow$ indices of the $Q$ smallest values of $\left\{ p_n^{(T)} \right\}_{n=1}^{N}$.

11: Query expert for labels in $\mathcal{Q}$.

12: **for** $n \in \mathcal{Q}$ **do**

13: $\quad y_n^{(T)} \leftarrow y_n$

14: **end for**

15: **for** $t = T + 1, T + 2, \ldots, T'$ **do**

16: $\quad \theta_{kv}^{(t)} = \dfrac{\sum_{n=1}^{N} \mathbb{I}\{y_n^{(t-1)} = k\} \cdot X_{nv}}{\sum_{n=1}^{N} \mathbb{I}\{y_n^{(t-1)} = k\} \cdot L_n}$

17: $\quad y_n^{(t)} \leftarrow \arg\max_{k \in [K]} \sum_{v \in [V]} X_{nv} \cdot \log(\theta_{kv}^{(t)})$

18: **end for**

19: **Output** $y_n^{(T')}$

---

Let $N$ be the number of unlabeled corpus documents and $K$ be the number of reference information documents. Recall that each label class has a single reference information document which may be a concatenation of many relevant documents. Let $V$ be the size of the vocabulary space, indexed $[V] = 1, 2, \ldots, V$. Throughout the paper, we will use $\Delta^V$ to denote the set of probability distributions over $[V]$ (i.e, a simplex in $V$ dimensions). Denote $X_{nv} \in \mathbb{N}$ as the count of word $v \in [V]$ in corpus document $n \in [N]$ and let $L_n := \sum_{v \in [V]} X_{nv}$ be the length of the $n$th corpus document. Similarly, denote $R_{kv}$ for the count of word $v$ in reference information document $k \in [K]$. In many parts of the implementation, it will be convenient to consider reference information as a distribution over the vocabulary space with non-zero entries. To do this, define the $r_{kv}$ as follows:

$$r_{kv} := \frac{R_{kv} + a}{\sum_{v \in [V]} (R_{kv} + a)} \tag{1}$$

This process is called *smoothing* and is commonly to convert counts into distributions with non-zero entries (Liu & Cheryl (2011)). For the real-world datasets, $a = 5$ is chosen, although any reasonable value of $a$ gave similar results. Throughout this paper, $X_n$ is used to denote the vector $\{X_{nv}\}_{v \in [V]}$ and $r_k$ denotes the probability distribution $\{r_{kv}\}_{v \in [V]}$.

### 3.2.2 Phase 2: Initial Label (Line 4).

Phase 2 generates an initial label $y_n^{(0)}$ for each corpus document $n \in [N]$ by finding the distribution $r_k$ which is *most likely to generate* the observed word counts $X_n$ from a Multinomial distribution with $L_n$ draws and probability vector $r_k$.

$$y_n^{(0)} = \arg\max_{k \in [K]} \binom{L_n}{X_{n1}, \dots, X_{nv}} \prod_{v=1}^{V} (r_{kv})^{X_{nv}} \tag{2}$$

$$= \text{argmax}_{k \in [K]} \sum_{v=1}^{V} X_{nv} \cdot \log(r_{kv}) \tag{3}$$

Since corpus documents and reference information documents are written in different contexts, often by different individuals, the initial label be far from perfect. Nevertheless, the empirical result show that the the initial label provides a reasonable estimate of the correct label.

An equivalent definition for $y_n^{(0)}$ is the class which minimizes the KL divergence (a measure of distance between distributions) between $x_n$ and $r_k$, where $x_n$ is a normalized and smoothed version of $X_n$ with any choice of $a$ (defined similarly to equation equation 1).

$$y_n^{(0)} := \text{argmin}_{k \in [K]} KL(x_n || r_k) \tag{4}$$

This equivalence is proven in Lemma E.1 in Appendix C

### 3.2.3 Phase 3: Iterated Label (Lines 5 to 8).

In Phase 3, the initial labels $y_n^{(0)}$ from from Phase 2 are iteratively updated by a supervised classification algorithm, in particular Multinomial Naive Bayes (MNB), a popular choice for text classification problems algorithms Kibriya et al. (2005); Frank & Bouckaert (2006); Abbas et al. (2019); Wang et al. (2015). Phase 3 runs for $T$ iterations, where $T$ is a user-defined parameter. For $t \in [T]$, train a MNB classifier while treating the labels $y_n^{(t-1)}$ as ground truth, then make predictions on the same data, updating some of the labels in the process. We state below the training and prediction process mathematically–see Appendix C.1 for background on the Multinomial Naive Bayes classifier.

- **Estimation.** Fitting a Multinomial Naive Bayes classifier for a $K-$class classification problem amounts to estimating $K$ probability distributions $\theta_k^{(t)} \in \Delta^V$ such that each distribution is the empirical average of all documents for which $y_n^{(t-1)}$ (label in the previous iteration) is $k$.

$$\theta_{kv}^{(t)} = \frac{\sum_{n=1}^{N} \mathbb{I}_{y_n^{(t-1)}=k} \cdot X_{nv}}{\sum_{n=1}^{N} \mathbb{I}_{y_n^{(t-1)}=k} \cdot L_n} \qquad k \in [K], v \in [V] \tag{5}$$

  At iteration $t$, the distribution $\theta_k^{(t)}$ serves as MNB's estimate for how documents with label $k$ are generated, according to the labels $y_n^{(t-1)}$ from the previous iteration.

- **Prediction.** For each corpus document $n$, predict a new label $y_n^{(t)}$ in the following equation equation 6:

$$y_n^{(t)} = \arg\max_{k \in [K]} \sum_{v \in [V]} X_{nv} \cdot \log(\theta_{kv}^{(t)}) \qquad n \in [N] \tag{6}$$

  Equation equation 5 is similar to equation 3 as MNB predicts the label $k$ which maximizes the likelihood of generating $X_n$ from $\theta_k$.

This iterative estimation and prediction process is Lines 5 to 8 of Algorithm 1. This process is repeated for each $t \in [T]$.. The output of Phase 3 is the label $y_n^{(T)}$, the iterated label before querying an expert in Phase 4.

| Documents | $N$ | $K$ | $V$ | $L_n$ (Average) | Reference Info | Length |
|-----------|-----|-----|-----|-----------------|----------------|--------|
| Pharma | 563 | 5 | 4334 | 102 | Standard Operating Procedures | 4731 |
| BBC | 1490 | 5 | 2094 | 119 | Wikipedia Articles | 3724 |
| NewsGroup | 5859 | 10 | 4971 | 96 | Wikipedia Articles | 5553 |

Table 2: Basic information regarding the three datasets

### 3.2.4 Phase 4: Expert Labels (Lines 9 to 18).

Let $Q$ be the available number of queries to an expert. If $Q = 0$, then we return $y_n^{(T)}$ as the final predicted labels from DCRI. IF $Q > 0$, then DCRI asks an SME to label $Q$ corpus documents chosen based on the probabilities generated by MNB.As with most classifiers, MNB not only provides a predicted label $y_n^{(T)}$ but also an associated probability vector $p_n$ over the $K$ classes. This probability is calculated by applying Bayes Rule to calculate the *posterior* probability of each label class given the observed word count, assuming a uniform prior. The formula is provided in equation equation 24 in Appendix C.1. We interpret $p_n$ as the *confidence* of MNB on the label $y^{(T)}$, where larger values of $p_n$ represent more confident predictions. Let $\mathcal{Q} \subset [N]$ be the set containing the $Q$ documents with the smallest $p_n$'s. SMEs are asked to review these $Q$ corpus documents, updating $y_n^{(T)}$ (the predicted label) to the correct label if the label was not correct already. Finally, given that some labels have been corrected, DCRI performs a few more iterative phases from $t = T + 1$ to $t = T'$, where $T'$ is a user-defined parameter. This iterative process is shown in Lines 15 to 18 and is similar to that of Phase 3. In practice, it is rare that labels changes during this second iterative phase.

### 3.2.5 Modularity.

While this work studies only the DCRI with MNB algorithm, the DCRI framework is modular to support many other approaches to each of the 4 phases. For example, the Phase 1 vectorization phase could be replaced with deep learning based language embeddings (e.g., BERT Devlin et al. (2019), GloVe Pennington et al. (2014), Word2Vec Mikolov et al. (2013)), which have become increasingly popular in recent years. Other supervised classification algorithms can be used for Phase 3, while other methods of prioritization can be used in Phase 4. Nevertheless, the following section shows that DCRI with MNB exhibits strong performance on real-world dataset, despite its simplicity. The strong interpretability of the BoW embedding and MNB classifier allows us to explain why DCRI with MNB performs well, which inspired the modeling presented in Section 5.

## 4 Evaluating DCRI with MNB on Real-World Datasets

This section presents the results of DCRI with MNB on 3 real-world datasets: one with deviations reports from a major pharmaceutical company and two public datasets related to news topic classification.

### 4.1 Datasets Overview

Table 2 contains basic information about the three real-world datasets we use in this paper.

### 4.1.1 Pharmaceutical Dataset.

The pharmaceutical dataset consists of 563 non-conformance records that occurred within the production bioreactor at one specific manufacturing site between 2011 and 2016. As mentioned in Section 1.1, the goal is to label each document with one of 5 *process steps* within the production bioreactor. We use Standard Operating Procedures (SOP)'s as reference information. Each process step's reference information consists of 1 or more lengthy reference information documents that describe how an technician should operate or

troubleshoot that particular process step. Our industry partners generously hand-labeled all the corpus documents with one of the 5 labels so that we can evaluate DCRI's performance.

**Public Datasets.** The other two datasets used to validate our methodology are public datasets with related to topic classification of news discussion and articles. The 20 Newsgroup dataset (hereby called Newsgroup) is a famous dataset consisting of news discussion posts from a pre-2000 website. For testing, we take a subset of 10 of the 20 topics, with each topic containing on average 585 corpus documents. The British Broadcasting Corporation (BBC) dataset is a public data set of over 1500 news snippets from BBC news articles with topics in one of five categories: technology, entertainment, politics, sports and technology.

A common theme to all three datasets is that the corpus documents are much shorter in length than the reference information and that the reference information are from a completely different source than the corpus documents themselves. While the two news datasets are balanced (over 5 or 10 classes), we note that the pharmaceutical deviations dataset is highly imbalanced. Since the bioreactor spends a non-uniform amount of time in each stage of the process, the number of deviations that occur in each stage is not uniformly distributed.

**Computing Environment.** All experiments in this paper were run locally on a 2019 Macbook Pro computer. The entire DCRI with MNB pipeline takes less than 10 seconds to complete.

The rest of this section presents empirical results for these three datasets in the case of $Q = 0$ (no expert labeling capacity) and $Q > 0$ (some expert labels available). For $Q = 0$, Section 4.2 shows that that DCRI with MNB is an effective way of quickly obtaining high-quality labels that significantly outperform unsupervised methods and closely approaches performance of supervised approaches. For $Q > 0$, Section 4.3 show that DCRI with MNB can better prioritize which $Q$ documents an expert should label compared to semi-supervised approaches that choose these documents at randomly.

### 4.2 Performance with No Expert Labels ($Q = 0$)

When $Q = 0$, DCRI relies entirely on the reference information to generate initial labels $y_n^{(0)}$ (Phase 2) and iteratively updates them to $y_n^{(T)}$ (Phase 3). Figure 4(a) shows the multi-class accuracy of various approaches described below.

- "DCRI Phase 2" and "DCRI Phase 3" show the accuracy of the labels $y_n^{(0)}$ and labels $y_n^{(T)}$, respectively.

- "Unsupervised LDA" shows the accuracy of an unsupervised Latent Dirichlet Allocation (LDA), where the accuracy is computed by taking the matching of clusters to labels that produces the *best* accuracy.

- "Supervised MNB" shows the out-of-sample accuracy of a supervised MNB classifier is trained on a random 80% of the training data with the true labels, and evaluated on the remaining 30%. This out-of-sample accuracy represents the accuracy that could achieve if a large number of labeled training examples were available.

- "Labels Saved vs. Semi-Supervised" compares DCRI with MNB with a semi-supervised benchmark where $Q > 0$ documents are chosen at random to be labeled, after which a classifier is used to generate labels for the remaining $N - Q$ documents. Then, an iterative supervised classifier similar to Phase 3 of DCRI is used to update these $N - Q$ labels iteratively for a fixed number of iterations. Such an approach, when using Naive Bayes as the supervised classifier, is referred to as Semi-supervised Naive Bayes Sristy & Somayajulu (2012); Zhao et al. (2016).

We highlight some of the insights from Figure 4(a).

**DCRI Phase 2 labels are relatively strong despite the fact that reference information come from an entirely different source than the corpus documents themselves (first column of Figure 4(a)**. This can be attributed to the presence of words whose frequency in the reference information is *consistent* with that of the documents. For example, the word `game` appears more frequently in both BBC articles with

| Dataset | DCRI Phase 2 (Initial) | DCRI Phase 3 (Iterated) | Unsupervised LDA | Supervised MNB | Labels Saved vs. Semi-Supervised |
|---|---|---|---|---|---|
| Pharma | 0.813 | 0.889 | 0.526 | 0.897 | 220 (39%) |
| Newsgroup | 0.652 | 0.832 | 0.551 | 0.879 | 330 (5.6%) |
| BBC | 0.799 | 0.956 | 0.775 | 0.968 | 75 (5.0%) |

(a) **Accuracy for DCRI and Benchmarks.** DCRI with MNB's final labels (after Phase 3) outperform that of unsupervised approaches, comes very close to a supervised MNB classifier (with sufficient training data) and matches the performance of the semi-supervised approach for up to a few hundred documents.

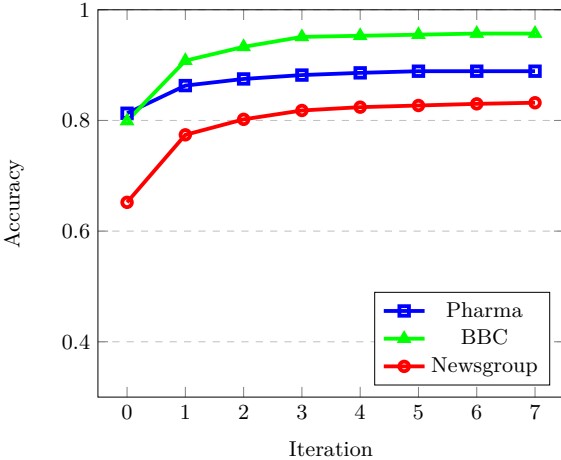

(b) **The Value of Iteration in Phase 3.** Iteratively training supervised classifiers on weak labels results in significant improvements in accuracy.

| | | Label Class | | | | |
|---|---|---|---|---|---|---|
| | | 1 | 2 | 3 | 4 | 5 |
| Proportion of Dataset | | 0.18 | 0.04 | 0.18 | 0.50 | 0.10 |
| DCRI Phase 2 (Initial Label) | Precision | 0.80 | 0.92 | 0.74 | 0.87 | 0.64 |
| | Recall | 0.91 | 0.96 | 0.75 | 0.77 | 0.90 |
| 100 Examples Uniformly at Random | Precision | 0.78 | 1.00 | 0.87 | 0.84 | 0.64 |
| | Recall | 0.93 | 0.17 | 0.65 | 0.97 | 0.55 |

(c) **Pharma Dataset: Precision and Recall Per Class.** DCRI produces labels that have better recall in the minority classes 2 and 5 (highlighted in red) than labels generated from the semi-supervised approach with 100 random examples.

Figure 4: DCRI with $Q = 0$ versus Benchmark Approaches

label `sports` and the Wikipedia articles on `sports`. On the other hand, there are also words for which the reference information provides a *corrupted* signal, such as the word `model` which occurs frequently in BBC `entertainment` articles (i.e., fashion model), but frequently in the Wikipedia article for `technology` (i.e., machine learning model). Finally, there are many words that are missing from the reference information altogether. For example, the word `Microsoft` appears often in BBC news articles on technology, but is not mentioned in the Wikipedia article for technology. Appendix D.1 provides more details on the breakdown of consistent, corrupt and missing words in the BBC dataset. This breakdown of vocabulary words is the basis of the Noisy Vocabulary Model in Section 5.

**The iterative process greatly improves performance between Phase 2 and Phase 3 labels (second column of Figure 4(a))**. Comparing the first and second columns of Figure 4(a), it is evident the iterative process in Phase 3 is valuable. In fact, Figure 4(b) shows how this accuracy changes in each iteration. This iterative process is useful because the supervised classifier can learn additional information about words not present in the reference information. For example, even though the word `Microsoft` was not useful in generating the initial label (since it does not appear in any of the Wikipedia reference information documents), the supervised algorithm can learn that `Microsoft` appears more often in documents whose initial label is assigned as `technology` over other classes. Thus, the supervised classifier learns that `Microsoft` is correlated with the `technology` class and can make better predictions on documents where `Microsoft` appears. Similarly, for words like `model` that were corrupt, the iterative process is able to correct that mistake. Appendix D.2 provides a deeper analysis of why this iterative process is useful.

**DCRI significantly outperforms unsupervised approaches (third column of Figure 4(a))** While unsupervised methods are great at detecting patterns within data, such methods fail to produce clusters that match a user's intentions and requires further expert labeling to split or combine clusters. This highlights the need for reference information, which when combined with a clustering-like approach results in clusters that are *anchored* to the desired classes. Appendix D.3 provides some examples of the clusters produced by LDA and how they fail to match up with the desired labeling scheme.

**DCRI's Phase 3 labels attain accuracy that is at least 94% of the supervised MNB's accuracy (last column of Figure 4(a))** Without any training examples at all, DCRI can achieve nearly the same performance if a large number of training examples were available, highlighting the significant value of reference information.

**DCRI with $Q = 0$ saves many labels over a semi-supervised approach.** Even with $Q = 0$, DCRI matches the performance of the semi-supervised approach with up to hundreds of labeled examples. In particular, DCRI saves more labels when classes are imbalanced. While the BBC and Newsgroup datasets are balanced across their 5 or 10 classes, the Pharma dataset is highly imbalanced as the bioreactor spends a non-uniform amount of time in each process step. A semi-supervised approach that randomly labels a subset of the data will disproportionally label examples from majority classes over minority classes. On the other hand, DCRI's initial label provides labels that are robust against class imbalance as the initial labels only depend on the reference information documents. Figure 4(c) highlights this phenomena in the Pharma dataset, which is highly imbalanced. While DCRI's initial labels from Phase 2 provide a consistent precision and recall across all 5 classes, a semi-supervised approach which randomly labels 100 documents and uses those labels to predict for the remaining documents achieves a poor precision recall on the minority classes (Class 2 and 5).

### 4.3 Performance with Some Expert Labels ($Q > 0$)

Figure 5(a) shows the accuracy of various approaches as a function of $Q$.

- "DCRI with Confidence" is DCRI with MNB where an expert reviews the $Q$ documents with the lowest confidence.

- "DCRI Random" is DCRI with MNB where an expert reviews a random subset of $Q$ documents.

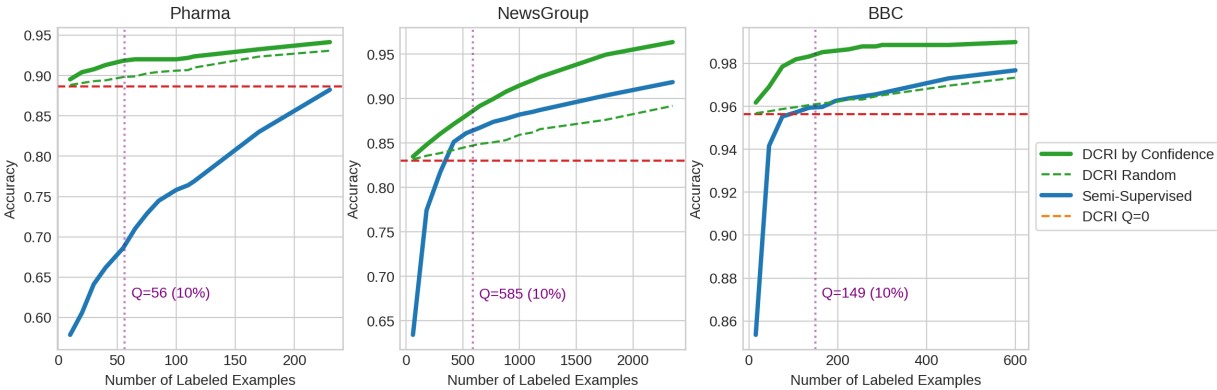

(a) **Accuracy while Varying** $Q$. DCRI with MNB (solid green line) outperforms the semi-supervised approach (solid blue line) when given the same $Q$ labeled examples.

| Dataset | DCRI by Confidence | DCRI Random | Semi-Supervised | DCRI with $Q = 0$ |
|---|---|---|---|---|
| Pharma | 0.925 | 0.901 | 0.692 | 0.889 |
| NewsGroup | 0.885 | 0.848 | 0.861 | 0.832 |
| BBC | 0.983 | 0.959 | 0.958 | 0.956 |

(b) **The Value of Reference Information.** At $Q = 0.10 \cdot N$, DCRI prioritized by confidence outperforms both DCRI with random labels and semi-supervised with random labels.

Figure 5: $Q > 0$: Results and Benchmarks

- **Semi-Supervised.** The semi-supervised Naive Bayes benchmark explained earlier in which $Q$ documents are randomly labeled, used to predict labels for the $N - Q$ remaining document,s then MNB is iteratively trained similar to Phase 3 of DCRI.

- "DCRI with $Q = 0$" shows the performance of DCRI with $Q = 0$ as a comparison, which clearly does not change as $Q$ increases.

**Prioritizing expert labels by confidence outperforms doing so randomly.** DCRI by Confidence always attains a higher accuracy than DCRI Random for any $Q$. For example, at $Q = 0.10 \cdot N$ (i.e., the labeling capacity is 10% of the corpus), Figure 5(b) shows that DCRI by Confidence effectively allocates its $Q$ labels to fix many more errors than DCRI Random, which randomly allocates such labels. For example, the error rate in the Pharma dataset decrease from $1 - 0.889 = 0.111$ to $1 - 0.925 = 0.075$, a 32% decrease, while reviewing randomly would only result in a 10% decrease in the error rate.

**Reference information is always useful regardless of $Q$.** For any $Q$, DCRI with Confidence outperforms the semi-supervised approach, highlighting the value of the DCRI approach regardless of the available labeling capacity. Having reference information allows DCRI to generate iterated labels whose confidences accurate capture which examples are most likely to be correct or incorrect. As a result, reference information indirectly leads to better prioritization of labeling capacity.

## 5 Data Generating Model: The Noisy Vocabulary Model

This section presents a generative model called the *Noisy Vocabulary Model* that is inspired by patterns in the real-world data discussed in Section 4, and aims to characterize how corpus documents and reference information documents are generated. Our modeling choices are inspired by empirical evidence from the real-world datasets. Section 5.1 presents an overview of the model, while Section 5.2 evaluates DCRI on synthetic data generated according to this model and presents new insights on the value of reference information.

### 5.1 Noisy Vocabulary Model

The DCRI framework relies on the observed word counts of the corpus documents $\{X_{nv}\}_{n\in[N],v\in[V]}$ as well as the empirical distributions of the reference information documents $\{r_k\}_{k=1}^K$. Figure 6(a) provides a schematic overview of the Noisy Vocabulary Model, which assumes that $\{X_{nv}\}_{n\in[N],v\in[V]}$ (item 3) and $\{r_k\}_{k=1}^K$ (item 5) are generated based on some unobserved latent parameters represented by items 1, 2 and 4.

**Generating Word Counts $\{X_{nv}\}_{n\in[N],v\in[V]}$ (Item 3).** We start by describing items 1, 2 and 3 at the top of Figure 6(a). It is assumed that the word counts $\{X_{nv}\}_{n\in[N],v\in[V]}$ (item 3) are drawn from a *mixture* model. Each of $N$ corpus documents are drawn independently according to the following steps.

1. Draw its label $y_n$ from a categorical distribution with probability given by $\gamma$, i.e. $P(y_n = k) = \gamma_k$.

2. Draw the length of the corpus document $L_n \sim \text{Pois}(L_0)$.

3. Draw the count of each word $X_{nv}$ with a multinomial distribution with $L_n$ trials and probability given by $\theta_{y_n}$, i.e.,

$$(X_{n1}, \ldots, X_{nv}) \mid y_n, L_n \sim \text{Multinomial}(L_n, \theta_{y_n})$$

Items 1 and 2 define a partition of the vocabulary space $\{\mathcal{D}_k\}$ that induces the set of distributions $\{\theta_k\}_{k=1}^K$ above. The set $\mathcal{D}_k$ contains vocabulary words which are *specific* to class $k$ in the corpus documents. A word is specific to a particular class if it occurs much more frequently in corpus documents belonging to that class than documents belonging to other classes. For example, in news topic classification, the words `baseball`, `fan` and `stadium` would be specific to the sports class while the words `television`, and `movie` would be specific to the entertainment class.

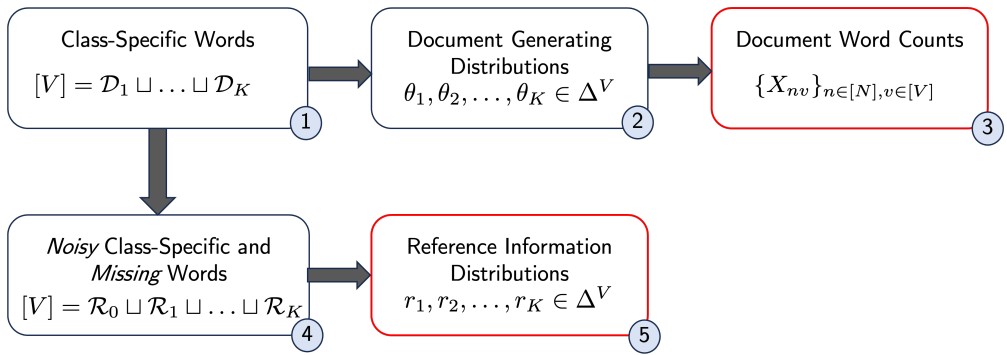

(a) **Data Generating Schematic** The noisy vocabulary model assumes that the the document word counts $\{X_{nv}\}_{n\in[N],v\in[V]}$ (item 3) are generated from distributions $\{\theta_k\}_{k=1}^K$ (item 2), which are based on a partitioning of the vocabulary space into $\{\mathcal{D}_k\}_{k=1}^K$ (item 1). A noisy estimate of $\{\theta_k\}_{k=1}^K$ is provided through $\{r_k\}_{k=1}^K$ (item 5) which is based on a different partitioning $\{\mathcal{R}_k\}_{k=0}^K$ (item 4).

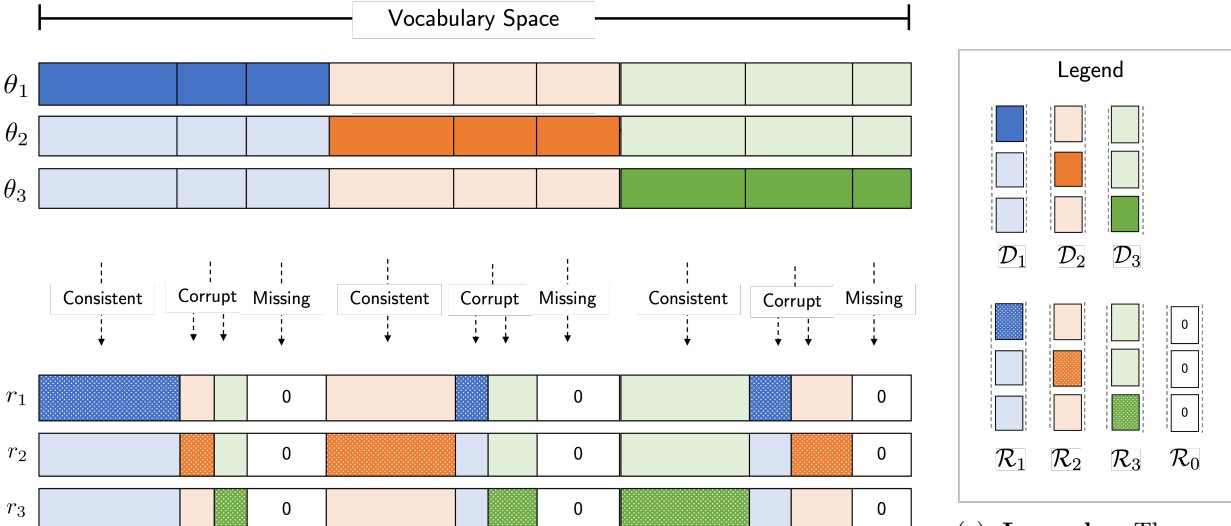

(b) **Example Heatmap of** $\theta_1, \theta_2, \theta_3$ **and** $r_1, r_2, r_3$. Darker colors represent larger numerical values. Noise causes some vocabulary words to be corrupt or missing, while other words stay consistent.

(c) **Legend.** The partitions $\mathcal{D}_k$ and $\mathcal{R}_k$ are depicted by the colored columns.

Figure 6: Noisy Vocabulary Model

This partitioning of vocabulary words $\{\mathcal{D}_k\}_{k=1}^{K}$ induces Document Generating Distributions $\{\theta_k\}_{k=1}^{K}$ (item 2) where $\theta_k$ is a distribution over the $[V]$. A large value of $\theta_{kv}$ indicates that word $v$ appears more frequently in documents belonging to class $k$. The distributions $\{\theta_k\}_{k=1}^{K}$ are constructed based on $\mathcal{D}_k$ in a way such that:

$$\text{If } v \in \mathcal{D}_k, \text{ then: } \theta_{kv} \gg \theta_{k'v} \forall k' \neq k \tag{7}$$

A word specific to class $k$ (i.e., $v \in \mathcal{D}_k$) has a higher probability of being generated in documents with $y_n = k$ than documents with $y_n \neq k$. A precise formula for $\{\theta_k\}_{k=1}^{K}$ given $\{\mathcal{D}\}_{k=1}^{K}$ is given in Appendix E.

Figure 6(b) provides an example for $K = 3$ where the distributions $(\theta_1, \theta_2, \theta_3)$ are visualized as heat maps over the vocabulary space with darker entries representing higher numerical values. As shown in the legend in Figure 6(c), the first one-third of the vocabulary space corresponds to $\mathcal{D}_1$, while the second one-third corresponds to $\mathcal{D}_2$, and last one-third corresponds to $\mathcal{D}_3$. Based on the colors, it is evident that equation equation 7 is satisfied.

**Generating** $\{r_k\}_{k=1}^{K}$ **(Item 5)** We now discuss items 4 and 5 in the bottom of Figure 6(a). Noise is added to the partitioning $\{\mathcal{D}_k\}_{k=1}^{K}$ (item 1) to generate a new partition $\{\mathcal{R}_k\}_{k=0}^{K}$ (item 4). Intuitively, $\mathcal{R}_k$ for $k \in [K]$ contain vocabulary words which are *specific* to class $k$ in the reference information (i.e., appear more frequently in that reference information document), while the additional set $\mathcal{R}_0$ contains words that do not appear in the reference information at all. That is, the reference information distributions $r_k$ are constructed such that

$$\text{If } v \in \mathcal{R}_k \text{ for } k \neq 0, \text{ then: } r_{kv} \gg r_{k'v} \qquad \forall k' \neq k \tag{8}$$
$$\text{If } v \in \mathcal{R}_0, \text{ then: } r_{kv} = 0 \qquad \forall k \in [K] \tag{9}$$

Comparing the partition $\{\mathcal{D}_k\}_{k=1}^{K}$ and $\{\mathcal{R}_k\}_{k=0}^{K}$ gives us consistent, corrupt and missing words, as defined in Section 4.2 when we discussed why reference information provide a reasonable initial label. These three types of words are defined mathematically in our model as follows.

- **Consistent.** $v \in \mathcal{D}_k$ and $v \in \mathcal{R}_k$

- **Corrupt.** $v \in \mathcal{D}_k$, but $v \in \mathcal{R}_{k'}$ with $k \neq k'$ and $k' \neq 0$

- **Missing.** $v \in \mathcal{D}_k$, but $v \in \mathcal{R}_0$

Figure 6(b) shows an example of consistent, corrupted and missing words. For $k \in [3]$, each vector $r_k$ is shaded strongly in the columns that correspond to the vocabulary words in $\mathcal{R}_k$, shaded lightly for the words in $\mathcal{R}_{k'}$ with $k' \neq k$ and $k' \neq 0$, and labeled with 0 for the words in $\mathcal{R}_0$. The set of consistent/good words are columns where the shading stays the same between $(\theta_1, \theta_2, \theta_3)$ and $(r_1, r_2, r_3)$; the set of corrupted words are ones where the shading changes color, while the set of missing words are columns where there is 0 in $(r_1, r_2, r_3)$.

Overall, the Noisy Vocabulary Model states that word counts are generated from a set of latent distributions $\{\theta_k\}_{k=1}^{K}$ which are based on a partitioning of the vocabulary space $\{\mathcal{D}_k\}_{k=1}^{K}$. However, DCRI is provided with a noisy estimate of $\{\theta_k\}_{k=1}^{K}$ given by $\{r_k\}_{k=1}^{K}$, which is derived from a different partitioning of the vocabulary space into $\{\mathcal{R}_k\}_{k=0}^{K}$. Mathematical details of this process can be found in Appendix E.

## 5.2 Simulation Study

This section presents a numerical study using synthetic data generated from the Noisy Vocabulary Model described in Section 5.1.

**Setup.** Our simulation study chooses $K = 5$ classes with $L_0 = 150$ and $\gamma_k = 1/5$ for all $k \in [5]$. The vocabulary space consists of $V = 2000$ words evenly distributed across $\mathcal{D}_1, \ldots, \mathcal{D}_5$ (i.e., $|D_k| = 400$). For each class $k$, out of these 400 words, a $p_G$ (good) fraction of them are consistent, a $p_C$ (corrupted) fraction of

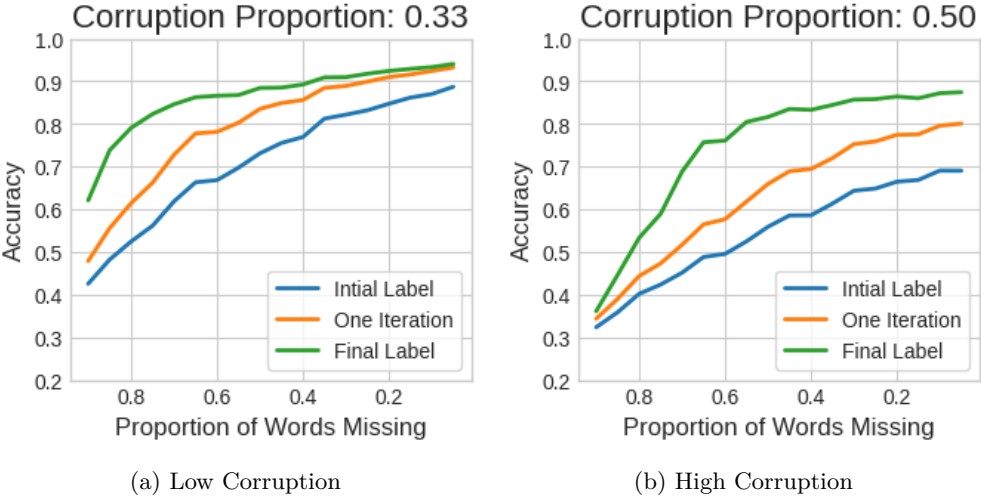

(a) Low Corruption            (b) High Corruption

Figure 7: Varying the Quality of Reference Information

them are corrupted to another class in a uniform way, and a $p_M$ (missing) fraction of them are missing from the reference info. That is:

$$|\mathcal{D}_k \cap \mathcal{R}_k| = p_G \cdot 400 \qquad |\mathcal{D}_k \cap \mathcal{R}_{k'}| = \frac{p_C \cdot 400}{K-1} \quad \forall k' \neq k \qquad |\mathcal{D}_k \cap \mathcal{R}_0| = p_M \cdot 400 \qquad (10)$$

For example, if $(p_G, p_C, p_M) = (1, 0, 0)$, then $\{\theta_k\}_{k=1}^K = \{r_k\}_{k=1}^K$ (i.e., perfect quality reference information) as all words are consistent. As $p_M$ or $p_C$ increases the quality of the reference information decreases. Our experiments will mainly vary the number of documents $N$ and the quality of the reference information through $(p_G, p_C, p_M)$. All experiments were run for a sufficiently large number of trials so that error bars are negligible. The experiments are organized to answer the following research questions.

- **Section 5.2.1**. How does the quality of reference information (measured by the amount of noise inserted into the document distributions) affect the performance of DCRI?

- **Section 5.2.2**. How does the size of the corpus and quality of reference information impact DCRI's edge over unsupervised approaches?

- **Section 5.2.3**. When does DCRI have the strongest advantage over semi-supervised approaches?

### 5.2.1 Impact of Reference Information Quality.

In this first study, we evaluate how the quality of reference information impacts the accuracy of the initial label from Phase 2 and iterated label from Phase 3 of DCRI. Assume that $Q = 0$ and fix $N = 500$. The quality of reference information is measured by two quantities: (1) $p_M$, the proportion of words missing and (2) the fraction of the remaining non-missing words that are corrupted, i.e. $\frac{p_C}{p_C + p_G}$, the corruption proportion. [1] Figure 7 shows the accuracy of the initial Phase 2 label, label after 1 iteration, and final iterated Phase 3 label, where the corruption proportion is fixed in each graph and the proportion of missing words $p_M$ is plotted on the $x$-axis. The left graph represents a setting where the corruption proportion is relatively low, while the right is the opposite. Decreasing the proportion of missing words (i.e., increasing the quality of the reference information) leads to a non-linear improvement in the accuracy of the iterated Phase 3 label for both low corruption and high corruption. Furthermore, the benefit is larger for high corruption. Such observations suggests that **a little bit of reference information quality can go a long way.** As a practitioner, spending a little more time to gather reference information that is closer in context to the unlabeled documents can have great benefits.

---

[1]For example, $\frac{p_C}{p_C + p_G} = 0.33$ means that for every 3 words that are consistent, 1 is corrupted to appear specific in a different class.

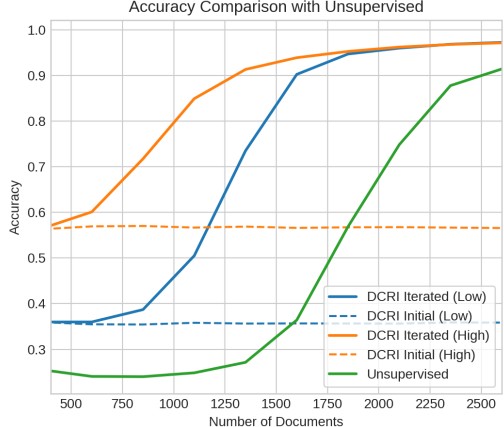

(a) **Comparison with Unsupervised.** For each $N$, the solid lines show the accuracies of DCRI with low and high quality as well as the unsupervised approach. The dashed lines show DCRI's initial label accuracy, which does not change with $N$.

| $N$ | DCRI vs. Unsupervsied | |
|---|---|---|
| | Low Quality | High Quality |
| 250 | 1.40 | 1.88 |
| 500 | 1.49 | 2.50 |
| 1000 | 2.03 | 3.41 |
| 1500 | 2.47 | 2.57 |
| 2000 | 1.28 | 1.29 |
| 2500 | 1.06 | 1.06 |

(b) **DCRI's Advantage** For each $N$, DCRI with low quality and high quality reference information achieves an accuracy that is between 1.06 and 3.41 times that of the unsupervised approach.

Figure 8: DCRI vs. Unsupervised while varying $N$

### 5.2.2 Impact of Corpus Size $N$ against Unsupervised Approaches

This experiment evaluates DCRI against unsupervised approaches while varying the number of corpus documents $N$. Similar to our empirical results, the accuracy of unsupervised algorithms is computed by taking the *best* accuracy across all permutations of the underlying classes to the clusters. Figure 8 shows the results of this experiment. Fix $p_M = 0.75$ and test two scenarios: $\frac{p_C}{p_C+p_G} = 0.66$ for low quality reference information and $\frac{p_C}{p_C+p_G} = 0.33$ for high quality. Figure 8(b) plots the ratio between DCRI's accuracy and that of the unsupervised approach. The insights are as follows.

**Reference information reduces the number of documents needed to achieve high accuracy compared to unsupervised approaches**. Consider the number of documents needed to achieve an accuracy of say 90%. The unsupervised approach (green line) takes roughly 2500 documents, whereas DCRI with low quality reference information (blue) takes about 1500 and DCRI with high quality reference information (orange) takes about 1250. This highlights that having a warm start (from the Phase 2 initial labels) from reference information significantly reduces the number of documents needed to achieve the same accuracy. Since documents from the Noisy Vocabulary Model are truly generated from a mixture of 5 distributions, as the number of documents increases, it becomes easier for an unsupervised algorithm to discover this underlying pattern. However, a large number of documents is needed before the unsupervised approach achieves a high accuracy. On the other hand, by having weak initial labels from reference information, DCRI is able to achieve strong accuracy with much fewer documents. However, **This benefit of reference information is most prominent when the number of documents is moderate.** Figure 8(b) shows that when $N$ is large, unsupervised approaches perform well already and reference information result in little improvement. On the other hand, when $N$ is small, reference information is helpful because it provides an initial label better than random guessing, but accuracy does not improve through iteration. Reference information is most useful over unsupervised approaches when $N$ is moderate: there are enough documents that the iterative process leads to an an improvement, but the not so many documents that the underlying clusters are easy to detect with the semi-supervised approach.

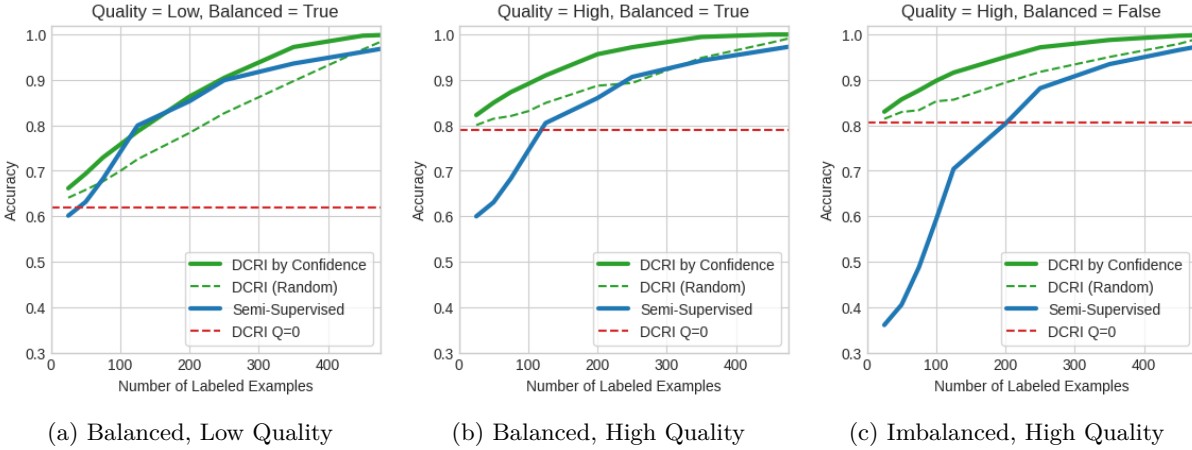

(a) Balanced, Low Quality      (b) Balanced, High Quality      (c) Imbalanced, High Quality

Figure 9: Comparison against Semi-Supervised Approaches.

### 5.2.3   Impact of Number of Labeled Examples $Q$ vs. Semi-Supervised Approach.

This section studies the case where some expert labeling capacity is available and compares DCRI with semi-supervised approaches. Fix $N = 500$ and vary $Q$, the number of labeled examples available from 25 to 475. We also examine combinations of the following two parameters.

- **Reference Information Quality**: High and Low, corresponding to $p_M \in \{0.40, 0.70\}$ and fix $p_C/(p_C + p_G)$ at 0.50.

- **Data Imbalance**: $\gamma = [0.2, 0.2, 0.2, 0.2, 0.2]$ for balanced and $\gamma = [0.35, 0.35, 0.1, 0.1, 0.1]$ for imbalanced.

Figure 9 compares the same 4 approaches as in Section 4.3 for combinations of {balanced, imbalanced} and {high quality, low quality}. In all 3 graphs, DCRI with Confidence outperforms DCRI Random, highlighting again the value of reviewing labels based on confidence over randomly. Figures 9(a) and 9(b) reinforce the observation that DCRI with Confidence always outperforms the semi-supervised approach, even when the quality of the reference information is low. Finally, comparing Figure 9(b) and 9 highlights the same observation as in Section 4.2, that semi-supervised approaches suffer in imbalanced datasets, while DCRI is unaffected.

## 6   Theoretical Property: Iteration improves Accuracy

A key element of the DCRI MNB methodology is the iterative process that improves the quality of the initial labels. This section provides a theoretical result for a special case of the generative model from Section 5.1 in which we *prove* that iterating the initial label results in an improvement in accuracy.

**Setup.** We consider a case with $K = 2$ classes, denoted $A$ and $B$ (this is to avoid confusion in notation later) Figure 10(a) depicts the distributions $\theta_A, \theta_B, r_A$ and $r_B$. The vocabulary space consists of $V = 2 \cdot (G + C + M)$ of which $|\mathcal{D}_A| = |\mathcal{D}_B| = G + C + M$. The corruption happens so that $G$ out of the $G + C + M$ words in $\mathcal{D}_A$ are consistent, $C$ are corrupt and $M$ are missing (likewise for $\mathcal{D}_B$). Denote the following sets of words:

$$\mathcal{V}_{AA} = \mathcal{D}_A \cap \mathcal{R}_A \qquad \mathcal{V}_{BB} = \mathcal{D}_B \cap \mathcal{R}_B \qquad |\mathcal{V}_{AA}| = |\mathcal{V}_{BB}| = G$$
$$\mathcal{V}_{AB} = \mathcal{D}_A \cap \mathcal{R}_B \qquad \mathcal{V}_{BA} = \mathcal{D}_B \cap \mathcal{R}_A \qquad |\mathcal{V}_{AB}| = |\mathcal{V}_{BA}| = C$$
$$\mathcal{V}_{A0} = \mathcal{D}_A \cap \mathcal{R}_0 \qquad \mathcal{V}_{B0} = \mathcal{D}_B \cap \mathcal{R}_0 \qquad |\mathcal{V}_{A0}| = |\mathcal{V}_{B0}| = M$$

Here, $\mathcal{V}_{AA}$ contain consistent words specific to class $A$, $\mathcal{V}_{AB}$ contains corrupt words specific to class $A$ in the documents but specific to class $B$ in the reference information...etc. The specific values of $\theta_A, \theta_B, r_A, r_B$

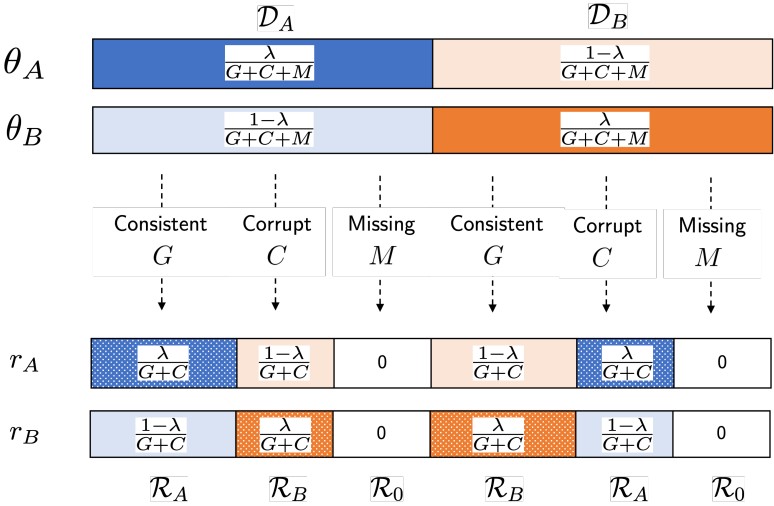

(a) **Setup.** $\theta_A, \theta_B, r_A, r_B$ are depicted above. There are $2 \cdot (G+C+M)$ vocabulary words which are partitioned so that $|\mathcal{D}_A| = \mathcal{D}_B = G + C + M$. Out of the words in $\mathcal{D}_A, \mathcal{D}_B$, a total of $G$, $C$ and $M$ words are consistent, corrupt or missing respectively. The values of $\theta_1, \theta_2, r_1, r_2$ are shown above as well.

| | $X_{A,A}$ | $X_{A,B}$ | $X_{A,0}$ | $X_{B,B}$ | $X_{B,A}$ | $X_{B,0}$ |
|---|---|---|---|---|---|---|
| $\log\left(\frac{r_A}{r_B}\right)$ | $+1$ | $-1$ | $0$ | $-1$ | $+1$ | $0$ |
| $\log\left(\frac{\theta_A^{(1)}}{\theta_B^{(1)}}\right)$ | $+1$ | $\alpha$ | $\beta$ | $-1$ | $-\alpha$ | $-\beta$ |
| $\log\left(\frac{\theta_A}{\theta_B}\right)$ | $+1$ | $+1$ | $+1$ | $-1$ | $-1$ | $-1$ |

(b) **Proof Sketch: Decision Boundary Coefficients.** Using the distributions $(r_A, r_B)$, $(\theta_A^{(1)}, \theta_B^{(1)})$ or $(\theta_A, \theta_B)$, construct a decision boundary between class $A$ and $B$. The coefficients of that decision boundary is shown above, where the numbers in each column are the coefficients of the variables $X_{AA}, \dots, X_{B0}$.

Figure 10: Special Case with Two Label Classes Denoted $A$ and $B$

(derived from the full model description in Appendix E) are shown in Figure 10(a) and the equations below.

$$\theta_{A,v} := \begin{cases} \frac{\lambda}{G+C+M} & v \in \mathcal{D}_A \\ \frac{1-\lambda}{G+C+M} & v \in \mathcal{D}_B \end{cases} \qquad\qquad \theta_{B,v} := \begin{cases} \frac{\lambda}{G+C+M} & v \in \mathcal{D}_A \\ \frac{1-\lambda}{G+C+M} & v \in \mathcal{D}_B \end{cases}$$

$$r_{A,v} := \begin{cases} \frac{1-\lambda}{G+C} & v \in \mathcal{R}_A \\ \frac{\lambda}{G+C} & v \in \mathcal{R}_B \\ 0 & v \in \mathcal{R}_0 \end{cases} \qquad\qquad r_{B,v} := \begin{cases} \frac{1-\lambda}{G+C} & v \in \mathcal{R}_A \\ \frac{\lambda}{G+C} & v \in \mathcal{R}_B \\ 0 & v \in \mathcal{R}_0 \end{cases}$$

Here, $\lambda > \frac{1}{2}$ is a parameter of the model; Theorem 6.1 holds for any choice of $\lambda > \frac{1}{2}$.

We define the following types of labels:

- **True Label.** $Y_n \in \{A, B\}$ is a random variable for the true label of document $n$. Recall from Noisy Vocabulary Model that in order to draw the word counts $X_{nv}$, we first its label $Y_n$.

- **Initial Label.** $Y_n^{(0)} \in \{A, B\}$ is a random variable for the Phase 2 initial label of document $n$. $Y_n^{(0)}$ is computed using the reference information distributions $r_A, r_B$.

- **Single Iteration Label.** $Y_n^{(1)} \in \{A, B\}$ is a random variable for the Phase 3 label of document $n$ after a *single* iteration. $Y_n^{(1)}$ is computed using $\theta_A^{(1)}$ and $\theta_B^{(1)}$, which are estimated from $Y_n^{(0)}$ in equation equation 5.

- **Optimal Label.** $Y_n^* \in \{A, B\}$ is the label assigned to document $n$ if $\theta_A, \theta_B$ were known. $Y_n^* = A$ if the word counts $X_{nv}$ are more likely to be drawn from $\theta_A$ than $\theta_B$. [2] This label will be used to develop intuition for the theorem.

We present our main theorem regarding the accuracy of the initial labels $Y_n^{(0)}$ and label $Y_n^{(1)}$ after a single iteration of DCRI.

**Theorem 6.1** (The Value of Iteration). *In the limiting case when $N \to \infty$, the accuracy of the iterated labels $\{Y_n^{(1)}\}_{n=1}^N$ is at least as high as the accuracy of the initial labels $\{Y_n^{(0)}\}_{n=1}^N$*

$$\lim_{N \to \infty} \frac{1}{N} \sum_{n=1}^N \mathbf{1}\{Y_n = Y_n^{(0)}\} \leq \lim_{N \to \infty} \frac{1}{N} \sum_{n=1}^N \mathbf{1}\{Y_n = Y_n^{(1)}\} \tag{11}$$

*By the law of large numbers (since each corpus document $n \in [N]$ is drawn independently), the above is equivalent to:*

$$P(Y = Y^{(0)}) \leq P(Y = Y^{(1)}) \tag{12}$$

*where $Y, Y^{(0)}$ and $Y^{(1)}$ are labels for an arbitrary corpus document drawn according to the generative model.*

A proof sketch of Theorem 6.1 is provided below. The formal lemmas are stated and proven in Appendix F. This proof has 4 major steps:

1. Lemma F.1: Characterize the decision boundary for labels $Y^{(0)}$ in terms of $(r_A, r_B)$

2. Lemma F.2: Characterize the decision boundary for $Y^{(1)}$ by defining $\theta_A^{(1)}, \theta_B^{(1)}$, which are computed based on $Y^{(0)}$ (a random variable). Note that $\theta_A^{(1)}, \theta_B^{(1)}$ are random variables, but converge as $N \to \infty$.

3. Lemma F.3, F.4 and F.5: Describe how the decision boundary changes between $Y^{(0)}$ and $Y^{(1)}$, in particular that $\alpha \geq -1$ and $\beta > 0$ (defined in equation equation 14 below).

4. Lemma F.6: Argue that such changes result in labels $Y^{(1)}$ that are at least as accurate as $Y^{(0)}$.

---

[2]Note that $Y_n^*$ may not equal $Y_n$ as a document can be drawn from $\theta_A$, but by chance $X_{nv}$ could be drawn so that it is "closer" to $\theta_B$.

We provide some brief intuition for the Lemmas above. For notational convenience, we omit the subscript $n$ when discussing an arbitrary document $n \in [N]$. Define $X_{AA} = \sum_{v \in \mathcal{V}_{AA}} X_v$ and equivalently for each of the sets $\mathcal{V}_{AA}, \ldots, \mathcal{V}_{B0}$. Let $Y^*$ be the *best* guess of label $Y_n^*$ if the latent $(\theta_A, \theta_B)$ were known exactly. Figure 10(b) depicts the decision boundaries for $Y^{(0)}, Y^{(1)}$ and $Y^*$ shown below:

$$X_{AA} - X_{BB} + {\color{red}-1} \cdot (X_{AB} - X_{BA}) + 0 + {\color{red}0} \cdot (X_{A0} - X_{B0}) > 0 \quad \Leftrightarrow \quad Y^{(0)} = A \tag{13}$$

$$X_{AA} - X_{BB} + {\color{red}\alpha} \cdot (X_{AB} - X_{BA}) + {\color{red}\beta} \cdot (X_{A0} - X_{B0}) > 0 \quad \Leftrightarrow \quad Y^{(1)} = A \tag{14}$$

$$X_{AA} - X_{BB} + {\color{red}1} \cdot (X_{AB} - X_{BA}) + {\color{red}1} \cdot (X_{A0} - X_{B0}) > 0 \quad \Leftrightarrow \quad Y^* = A \tag{15}$$

To see why $Y^{(1)}$ is more accurate than $Y^{(0)}$, notice how $Y^{(0)}$ is different from $Y^*$. While $Y^*$ places weight of $+1$ on $X_{AB} - X_{BA}$ and $X_{A0} - X_{B0}$, $Y^{(0)}$ places a weight of $-1$ on $X_{AB} - X_{BA}$ and a weight of $0$ on $X_{A0} - X_{B0}$. The $-1$ is due to the corruption (e.g., words in $X_{AB}$ appear to be specific to class $B$ in the reference information) while the $0$ is due to missing words. The core idea of our proof is to show that $Y^{(1)}$ is an improvement over $Y^{(0)}$ in that it puts a weight of $\alpha > -1$ on $X_{AB} - X_{BA}$ and a weight of $\beta > 0$ on $X_{A0} - X_{B0}$, effectively moving the decision boundary closer to $Y^*$.

## 7 Conclusion

This paper proposed a novel problem of classifying documents into a desired labeling scheme in a semi-automated way, without the need for excessive manual labeling. We developed the Document Classification with Reference Information (DCRI) paradigm and a specific implementation DCRI with MNB. DCRI leverages subject matter expertise to gather reference information, which are used to generate weak labels for each document quickly. When no manual labeling capacity is available, DCRI outperforms unsupervised approaches, while when some manual labeling capacity is available, DCRI outperforms semi-supervised approaches. To explain DCRI's strong performance, we developed a theoretical model in which reference information is modeled as a corrupted version of the documents. Through comprehensive numerical studies and a theoretical analysis, we showed insights on the value of reference information that paralleled those on the real-world dataset.

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

# A    Appendix

# B    Appendix to Section 3

## B.1    Bag of Words Vectorization

The bag of words vectorization process can be broken down into two main steps:

**Step 1. Text Cleaning and Tokenization.**    Given a corpus of documents, we first perform the following text cleaning tasks on each of the documents.

1. Convert all text to lower case.

2. Remove punctuation and special characters.

3. Remove stop words such as "and", "the" or "is"

4. Stemming / Lemmatization: convert words to their base or root form, for example "running" to "run".

Then, the remaining cleaned text is tokenized by splitting on white spaces. For example, the sentence "The cat sat on the Mat" would be cleaned to "cat sat mat" and tokenized into ['cat', 'sat', 'mat']. Using this tokenization, we can convert each document into a vector of counts of each token.

**Step 2. Filtering the Vocabulary.** Take all the tokens created from tokenizing each document and look at how many documents they appear in. Filter only for tokens that appear in at least 1% of documents, but no more than 50% of documents. Tokens that appear in less than 1% of the documents are too rare to have any meaningful impact in classification (e.g., a serial number or a person's name that only occurs in very few documents). On the other hand, tokens that occur in more than 50% of all documents may be stop words specific to this dataset. For example, the word "reporting" in news topic classification or "bioreactor" in process step classification.

After filtering for tokens that satisfy the above criteria, we obtain a final vocabulary space of $V$ tokens. Each document is represented by a count of tokens over this vocabulary space.

## C Equivalence of KL Divergence and MNB

Using the definition of KL Divergence, we can get:

$$\text{argmin}_{k \in [K]} D_{KL}(x_n || r_k) = \text{argmin}_{k \in [K]} \sum_{v \in [V]} x_{nv} \cdot \log \left( \frac{X_{nv}}{r_{kv}} \right) \tag{16}$$

$$= \text{argmax}_{k \in [K]} \sum_{v=1}^{V} X_{nv} \cdot \log(r_{kv}) \tag{17}$$

The first equality is by definition of KL divergence. The second equality is true by noticing that $X_{nv} \cdot \log(X_{nv})$ does not depend on $k$ and by negating the entire expression (turning argmin into argmax). Replacing $x_{nv}$ with $X_{nv}$ is valid since $X_{nv} = X_{nv} \cdot L_n$.

We choose the KL divergence as a measure of similarity due to its nice properties described in Lemma E.1 below.

**Lemma C.1** (KL Divergence Interpretation). *Given a set of $K$ probability distributions $\{r_k\}_{k=1}^{K}$ and any vector of positive integers $X_n$ (normalized as $x_n = \frac{X_{nv}}{\sum_{v \in [V]} X_{nv}}$) the following are equivalent:*

1. *The class $k$ which minimizes the KL divergence between $x_n$ and $r_k$.*

2. *The class $k$ for which a multinomial distribution with $L_n = \sum_{n=1}^{N} X_{nv}$ trials and probability vector $r_k$ has the highest probability of generating the counts $X_n$. This is equivalent to the class that a MNB predicts assuming $\theta_k = r_k$.*

*Proof.* Proof of Lemma E.1 For 1., by the definition of KL divergence:

$$\text{argmin}_{k \in [K]} KL(x_n || r_k) = \text{argmin}_{k \in [K]} \sum_{v \in [V]} x_{nv} \cdot \log \left( \frac{X_{nv}}{r_{kv}} \right)$$

$$= \text{argmax}_{k \in [K]} \sum_{v=1}^{V} X_{nv} \cdot \log(r_{kv})$$

For 2., write out the probability mass function of the Multinomial distribution as follows:

$$\arg\max_{k\in[K]} \binom{L_n}{X_{n1},\ldots,X_{nv}} \prod_{v=1}^{V}(r_{kv})^{X_{nv}} = \arg\max_{k\in[K]} \log \prod_{v=1}^{V}(r_{kv})^{X_{nv}} \tag{18}$$

$$= \mathrm{argmax}_{k\in[K]} \sum_{v=1}^{V} X_{nv} \cdot \log(r_{kv}) \tag{19}$$

This completes the proof $\qquad\qquad\qquad\qquad\square$ $\qquad\qquad\qquad\qquad\qquad\square$

## C.1  Multinomial Naive Bayes Background

Multinomial Naive Bayes is a popular classification algorithm commonly for supervised text classification (see e.g. Kibriya et al. (2005); Abbas et al. (2019); Wang et al. (2015); Frank & Bouckaert (2006)). MNB assumes that the word counts $X_{nv}$ for a document with a true label of $y \in [K]$ is drawn from a Multinomial distribution over $L_n := \sum_{v\in[V]} X_{nv}$ trials, with each trial drawn using the probability vector $\theta_y$ over the vocabulary space $[V]$.

Training a MNB model on data $(X_n, y_n)_{n=1}^{N}$ amounts to estimating $K$ probability distributions $\theta_k$, one for each label class. The $\theta_k$ are chosen to maximize the likelihood of seeing the data $(X_n, y_n)_{n=1}^{N}$, where the document counts $X_n$ are assumed to be drawn from $\theta_{y_n}$. $\theta_k$ is taken as the maximum likelihood estimator as follows:

$$(\theta_1,\ldots,\theta_K) = \mathrm{argmax}_{p_1,\ldots,p_K} \prod_{n=1}^{N} \binom{L_n}{X_{n1},\ldots,X_{nv}} \prod_{v=1}^{V}(p_{y_n,v})^{X_{nv}}$$

$$= \mathrm{argmax}_{p_1,\ldots,p_K} \sum_{n=1}^{N}\sum_{v=1}^{V} X_{nv} \cdot \log(p_{y_n,v}) \tag{20}$$

Here, we also constrain $p_1,\ldots,p_K$ to be valid probability vectors (e.g. non-negative and sum to 1).

A closed form for $\theta_k$ can be obtained by decomposing across each value of $k \in [K]$.

$$\theta_k = \arg\max_{p_k\in\Delta^V} \sum_{n=1}^{N} \mathbb{I}\{y_n = k\} \cdot X_{nv} \cdot \log(p_{kv}) \qquad k \in [K]$$

This can be readily verified from first order conditions that $\theta_{kv}$ must be proportional to $\sum_{n=1}^{N} \mathcal{I}\{y_n^{(t-1)} = k\} \cdot X_{nv}$ (i.e. the number of times word $v$ appears in documents with $y_n^{(t-1)} = k$), which gives the following expression for $\theta_k$

$$\theta_{kv} = \frac{\sum_{n=1}^{N} \mathbb{I}_{y_n=k} \cdot X_{nv}}{\sum_{n=1}^{N} \mathbb{I}_{y_n=k} \cdot L_n} \qquad k \in [K] \tag{21}$$

Once $\{\theta_k\}_{k=1}^{K}$ has been estimated from $(X_n, y_n)_{n=1}^{N}$, a prediction can be made for any word count vector $X_n$ by computing the $\theta_k$ most likely to generate $X_{nv}$:

$$y_n = \arg\max_{k\in[K]} \binom{L_n}{X_{n1},\ldots,X_{nv}} \prod_{v=1}^{V}(\theta_{kv})^{X_{nv}} = \arg\max_{k\in[K]} \sum_{v\in[V]} X_{nv} \cdot \log(\theta_{kv}) \tag{22}$$

Likewise, a probability for $y_n$ can be computed by using Bayes Rule with a uniform prior over the $K$ classes. The probabiilty $p_n$ is computed as:

(a) Corruption of Strong Words from Documents to Reference Information

| Word | Document Class | Reference Info Class |
|------|----------------|---------------------|
| `play` | Sport | Entertainment |
| `camera` | Entertainment | Technology |
| `model` | Business | Technology |
| `left` | Politics | Sports |
| `street` | Entertainment | Politics |
| `system` | Entertainment | Technology |
| `accused` | Politics | (missing) |
| `common` | Politics | (missing) |
| `Microsoft` | Technology | (missing) |
| `decline` | Business | (missing) |
| `fight` | Sport | (missing) |
| `praise` | Entertainment | (missing) |

(b) Examples of corrupt and Missing Words

Figure 11: Comparison between documents and reference information. Words that should indicate strength in particular class may be corrupt to indicate strength in a different class, or be entirely missing from the reference information altogether. Intensities may also differ between documents and reference information (not shown).

$$p_n := \frac{\binom{L_n}{X_{n1},\ldots,X_{nv}} \prod_{v=1}^{V} (\theta_{kv})^{X_{nv}}}{\sum_{k \in [K]} \binom{L_n}{X_{n1},\ldots,X_{nv}} \prod_{v=1}^{V} (\theta_{kv})^{X_{nv}}} \tag{23}$$

$$= \frac{\prod_{v=1}^{V} (\theta_{kv})^{X_{nv}}}{\sum_{k \in [K]} \prod_{v=1}^{V} (\theta_{kv})^{X_{nv}}} \tag{24}$$

## D  Appendix to Section 4: Empirical Results

### D.1  Initial Labels

Figure 5(b) shows that the DCRI initial label's accuracy is 0.813 and 0.799 for the 5-way classification problems (Pharma and BBC) and 0.652 for the 10-way classification (Newsgroup). This is an impressive classification accuracy (compared to the random guessing baselines of 0.20 and 0.10). In this subsection, we will seek to explain this accuracy by examining how the documents and reference information differ.

**True Distribution vs. Reference Information Distribution.** Using the true labels $y_n$ (which our DCRI methodology does not have access to), we can construct the "true" conditional distributions of each class defined as $(\theta_k^{doc})_{k=1}^{K}$ by training a MNB on these true labels, resulting in the conditional distributions.

$$\theta_k^{doc} = \frac{\sum_{n=1}^{N} \mathbb{I}_{y_n=k} \cdot X_n}{\sum_{n=1}^{N} \mathbb{I}_{y_n=k} \cdot L_n} \qquad k \in [K]$$

How does $\theta_k^{doc}$ compare to $r_k$? In other words, how does the true conditional distribution of the documents under the correct labels compare to the distributions we estimated from the reference information? We first define some key terminology used in our analysis.

**Definition 1.** *Given $K$ conditional probability distributions $(\theta_1, \ldots, \theta_K)$ where $\theta_k \in \Delta^{V-1}$ is the generative probability of each class (with $\theta_{kv} > 0$ for all $v \in [V]$), we define the following for each word $v \in [V]$:*

- *The class that $v$ is specific with respect to $\theta$ is defined as the class that maximizes the probability of generating $v$, i.e.*

$$\text{CAT}(v, \theta) = \arg\max_{k \in [K]} \log \theta_{kv} \qquad (25)$$

- *The **separation** of $v$ with respect to $\theta$ is defined as the difference between the top log proportion of $v$ and the second highest log proportion of $v$:*

$$\text{SEP}(v, \theta) = \max_{k \in [K]} \log \theta_{kv} - \max_{k \neq \text{CAT}(v,\theta)} \log \theta_{kv} \qquad (26)$$

- *We will informally say that a word $v$ is **strong** in class $k$ to mean that $\text{CAT}(v, \theta) = k$ and $\text{SEP}(v, \theta)$ is large.*

Words $v$ with a high $\text{SEP}(v, \theta^{doc})$ are ones that truly distinguish between classes in the documents as this implies that one class (i.e. documents from class $\text{CAT}(v, \theta^{doc})$ have a much higher proportion of this word than documents from other classes). Recall that a document is classified by Multinomial Naive Bayes with distributions $\theta$ using the formula:

$$\arg\max_{k \in [K]} \sum_{v=1}^{V} X_{nv} \cdot \log \theta_{kv}$$

A word $v$ with high $\text{SEP}(v, \theta)$ will contribute highly to the sum for class $k = \text{CAT}(v, \theta)$ in the above $\arg\max$ and not contribute highly to the classes $k \neq \text{CAT}(v, \theta)$.

Figure 11(a) shows how these words are represented in the reference information. For each row, we consider all words $v$ for which $\text{SEP}(v, \theta^{doc}) \geq 0.4$, i.e. words that appear at least $e^{0.4} \approx 1.5$ times more frequently in one class than the rest. This threshold is chosen somewhat arbitrarily for the sake of explaining our methodology. Such specific words are placed into one of 6 columns (5 for the classes and the last for "Weak/Missing"). For all such specific words, we say that the word is in specific in class $\text{CAT}(v, r)$ in the reference information if $\text{SEP}(v, r) \geq 0.4$. Otherwise, if $\text{SEP}(v, r) < 0.4$, then we say that word is weak or missing.

We make the following observations from Figure 11.

- **Missing Words.** A majority of words that have high separation in the documents are not present in the reference information or have very weak separation. This makes sense as words there are many words which may be used disporportately often in BBC articles of a certain class, but not be general words that appear in Wikipedia. For example, BBC articles belonging to the class Politics often use the word `common` as in the House of Commons (equivalent to the House of Representatives in the US), but this word does not appear in the Wikipedia article for politics as such an article discusses more broad topics than any one country's political system. Figure 11(b) shows some examples of such missing words.

- **Corrupt Words.** Out of the remaining words $v$ for which $\text{SEP}(v, \theta^{doc}) \geq 0.4$ and $\text{SEP}(v, r) \geq 0.4$ in both the documents and reference information, a majority of them belong to the correct class, i.e. $\text{CAT}(v, r) = \text{CAT}(v, \theta^{doc})$. For example, common words like There are certainly some corrupt words as well where $\text{CAT}(v, r) \neq \text{CAT}(v, \theta^{doc})$. For example, the word `plays` occurs often in BBC articles involving sports to refer to plays of a sports game, but it appears often in the Entertainment Wikipedia article. Another example is the word `model` which is used in BBC articles in the context of business model, whereas in Wikipedia this word often refers to computer models. Figure 11(b) shows some additional examples of such corrupt words.

The two insights above provide intuition as to why the initial labels provide a good estimate of the true label, but are still quite far from being perfect. For a document with true label $k$, it will most likely contain more words that belong to $k$ than words that belong to other categories. For a majority of these words, the reference information will give us no indication that such a word belongs to class $k$. For the minority of words that strong separability in the reference information, a majority of them will correct tell us that this word is indicative of class $k$. As a result, we will label this document as class $k$ with probability much better than random guessing, but not anywhere close to perfect.

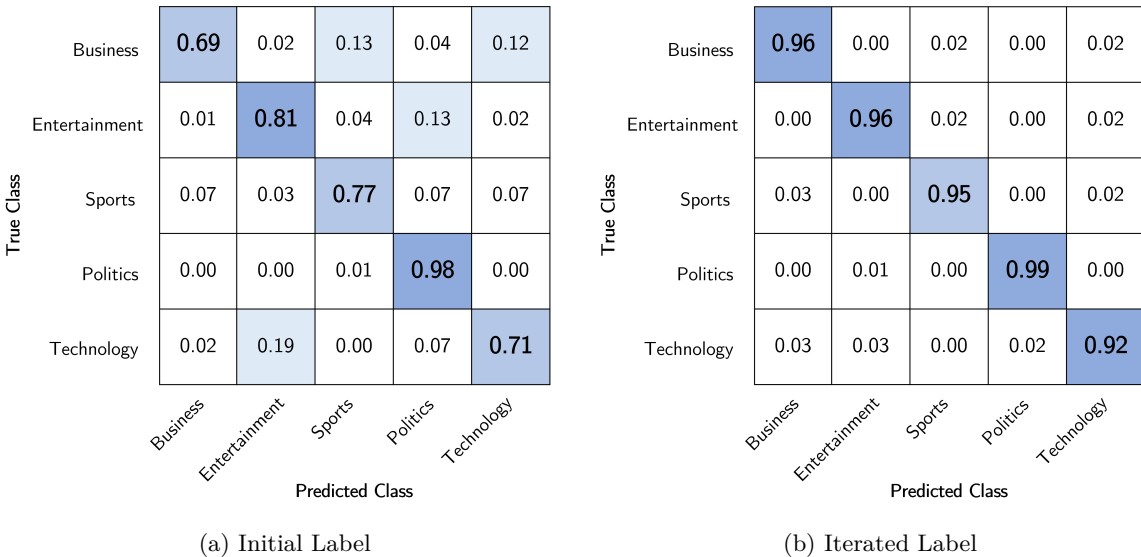

(a) Initial Label                                    (b) Iterated Label

Figure 12: Confusion matrices for $y_n^{(0)}$ (initial label) and $y_n^{(T)}$ (iterated label) with respect to $y_n$, the true labels.

## D.2    Iterated Labels

While the initial labels provided a reasonable estimate of the true labels, we saw in all three datasets that the iterative process resulted in a significant improvement in accuracy. Figure 12 shows the confusion matrix for $(y_n^{(0)})_{n=1}^N$ (initial labels) and final iterated labels $(y_n^{(T)})_{n=1}^N$. It is evident that many labels were corrected from the initial label to the final iterated label. But how and why does this happen? Consider a single iteration where the labels are updated from $(y_n^{(0)}))_{n=1}^N$ to $(y_n^{(1)})_{n=1}^N$. Recall that the initial labels $(y_n^{(0)})_{n=1}^N$ is generated using distributions $(r_k)_{k=1}^K$ gathered from the reference information, while $(y_n^{(1)})_{n=1}^N$ are generated from $\theta^{(1)}$, which is generated based on $(y_n^{(0)})_{n=1}^N$. We split our explanation into two steps: First, we explain why $(\theta_k^{(1)})_{k=1}^K$ is able to correct corrupt words and activate missing words from $(r_k)_{k=1}^K$. Then, we explain why such an improved $(\theta^{(1)})_{k=1}^K$ can result in labels being corrected from $(y_n^{(0)})_{n=1}^N$ to $(y_n^{(1)})_{n=1}^N$.

**Correcting Words from $r$ to $\theta^{(1)}$.** Recall from Section 4.2 that conditional distributions $(r_k)_{k=1}^K$ induced from the reference information has many mistakes, namely words that are corrupt and words that are missing. Consider a particular missing word such as `Microsoft`, which has a high separation under the technology class in the documents, but does not appear at all in the reference information (i.e. none of the Wikipedia articles mention Microsoft as a company explicitly). The fact that `Microsoft` has high separation means that `Microsoft` appears disproportionately often in documents of class technology relative to all other classes. Consider all the documents containing the word `Microsoft`. Since Microsoft is a specific word in the class technology, a strong majority of these documents have true class technology. Given that our initial labels $(y_n^{(0)})_{n=1}^N$ are reasonably accurate, a large number of these documents with `Microsoft` are assigned to the class technology. Therefore, when we estimate $\theta^{(1)}$, which is computed from $(y_n^{(0)})_{n=1}^N$, we will estimate $\theta_{\texttt{tech,Microsoft}}^{(1)}$ to be higher than the other classes. The bottom line is that documents whose initial label is technology are more likely to have a true label of technology (compared to documents with a different initial label). Since `Microsoft` appears more often in documents with true class of technology than other documents, this also results in `Microsoft` appearing more often in documents with initial label technology. In this way, even though `Microsoft` had no signal in the reference information, the fact that our initial labels are reasonably strong induced an iterated distribution $(\theta_k^{(1)})_{k=1}^K$ in which `Microsoft` is now specific in the technology class as desired. A similar reasoning can be applied to words that are corrupt.

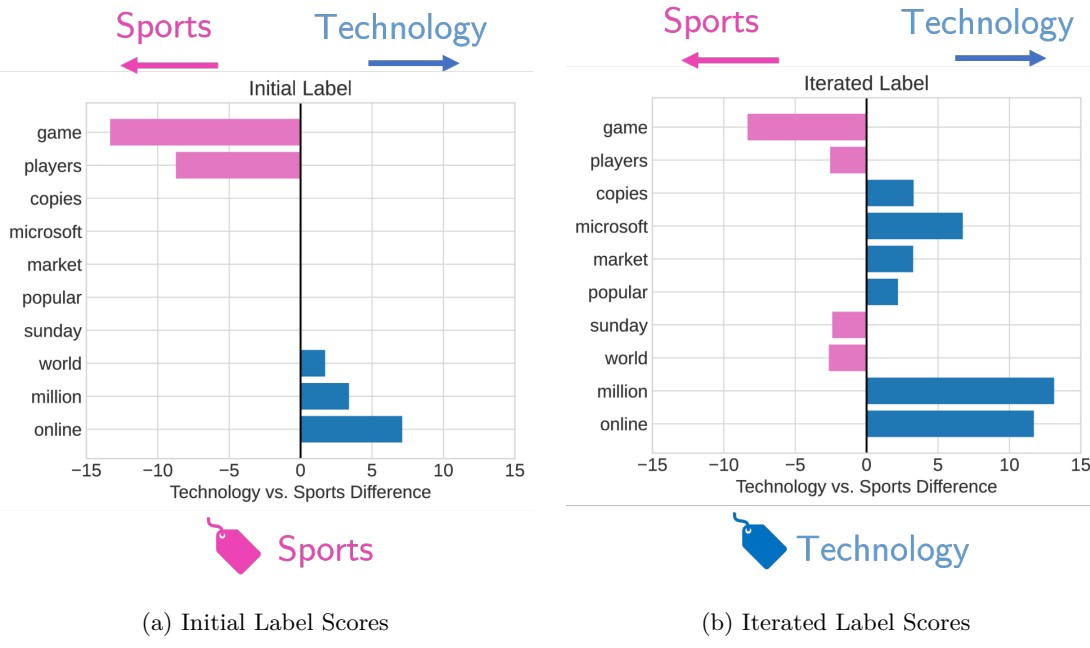

(a) Initial Label Scores                (b) Iterated Label Scores

Figure 13: Example of a document which was corrected after iteration.

**Correcting Labels Given A Better $\theta^{(1)}$.** We saw above that the iterative process can correct corrupt words and activate missing words. Here, we present a concrete example of how this correction can lead to a correction in the labels $(y_n^{(1)})_{n=1}^N$ from $(y_n^{(0)})_{n=1}^N$.

**Example 1.** *In this example, we demonstrate a document which had an initial label of sports but a label of technology after a single iteration. The example document we choose is one that talks about a popular multiplayer online game, which BBC classifies under the technology class (sports is reserved for traditional sports).*

*For this document $X_n$, we focus our attention on the 10 vocabulary words shown above and suppose that there are only two classes: sports and technology (tech). This document would be assigned an initial label of as sports if and only if:*

$$\sum_{v=1}^{10} X_{nv} \cdot \log r_{tech,v} > \sum_{v=1}^{10} X_{nv} \cdot \log r_{sports,v}$$

*On the left, we are showing $X_{nv} \cdot (\log r_{tech,v} - \log r_{sports,v})$ for each of these 10 vocabulary words $v$. This can be interpreted as the contribution of each word towards the class of technology (away from sports). The initial label of sports was assigned because the words game and players overpowered the words world, million and online. The remaining 5 words were not present in the reference information and thus provided no signal. On the other hand, after the iterated algorithm, the words copies, Microsoft, market, popular and sunday have been activated. In addition, the words million and online are now even specific in the technology class. This results in an iterated label of technology, thus correcting the initial label.*

## D.3 Comparison to Unsupervised Methods

As we saw from Figure 4, DCRI significantly outperforms purely unsupervised algorithms that do not leverage reference information at all. Unsupervised algorithms fail to produce high accuracy because it is difficult when clustering in high-dimensional space to guarantee that the clusters will exactly match a user's expectations, i.e. the exact $K$ classes we want. Figure 14 shows the result of the K-Means clustering algorithm on the BBC dataset. The table shows the number of documents with a particular true class (row) assigned to each of the

| | Cluster 1 | Cluster 2 | Cluster 3 | Cluster 4 | Cluster 5 |
|---|---|---|---|---|---|
| Business | 193 | 9 | 134 | 1 | 3 |
| Entertainment | 9 | 242 | 10 | 3 | 0 |
| Politics | 7 | 10 | 51 | 220 | 4 |
| Sports | 4 | 71 | 1 | 270 | 6 |
| Technology | 3 | 0 | 3 | 8 | 242 |

```
Cluster 1      Cluster 2      Cluster 3
  Shares         Film          Economic
  Firm           Awards         Growth
  Stake          Actor         Deficit
  Profits        Best          Dollar
  Company        Olympic        Rates

        Cluster 4      Cluster 5
         Election        Users
          Party         Mobile
         England       Software
         Scotland      Digital
         Chelsea       Microsoft
```

Figure 14: Unsupervised Algorithm Results. An unsupervised algorithm produces 5 clusters for the BBC dataset that do not match the underlying clusters well

5 clusters from K-Means (column). Here, we have already matched the clusters to the 5 classes to generate the highest accuracy (i.e. the sum of the entries on the diagonal is maximized over all permutations). On the right, the top 5 keywords for each of the clusters are shown. The words in black are words that belong to the class matched with this cluster, while the colored words are words that belong to a different class. We observe the following:

- **Good Clusters.** Clusters 1 and 5 contain documents which are very homogeneously from the technology and technology classes respectively. Although cluster 1 contains mostly technology documents, it sadly does not contain all technology documents as a substantial number of them are assigned to cluster 3.

- **Ambiguous Clusters.** Clusters 2, 3, and 4 do not correspond well with only a single class of documents. The top words from cluster 2 not only belong to the entertainment class, but also the sports class, namely words involving the Olympic games. Cluster 3 contains a large number of words related to economics, which belong to the technology class. Cluster 4 contain both documents involving politics and sports as there are large number of geographical words describing locations in Great Britain.

While unsupervised methods are great at detecting patterns within data, such methods fail to produce clusters that match a user's intentions and requires further expert labeling to split or combine clusters. This highlights the need for reference information, which when combined with a clustering-like approach results in clusters that are *anchored* to the desired classes, resulting in high accuracy.

## E   Appendix to Section 5: Generative Model

This section presents the mathematical details of the noisy vocabulary model described in Section 5.1. Inputs to the model include all the variables in the top portion of Table 3. Given these variables, the modeling process constructs $\theta_k$ and $\theta_k^{(0)}$ deterministically and $(X_n, y_n, L_n)_{n=1}^N$ in a stochastic fashion.

**Corpus Documents.**   Each of $N$ corpus documents are drawn independently according to the following steps.

1. Draw its true label $y_n$ from a categorical distribution with probability given by $\gamma$, i.e. $P(y_n = k) = \gamma_k$.

2. Draw the length of the corpus document $L_n \sim \text{Pois}(L_0)$.

| Symbol | Meaning |
|---|---|
| $N \in \mathbb{N}$ | Number of corpus documents, indexed $1, 2, \ldots, N$ |
| $K \in \mathbb{N}$ | Number of classes, indexed $1, 2, \ldots, K$ |
| $V \in \mathbb{N}$ | Number of vocabulary words, indexed by $1, 2, \ldots, V$ |
| $L_0 \in \mathbb{N}$ | Mean length of each corpus document |
| $\gamma \in \Delta^{K-1}$ | Distribution of true labels for the corpus documents |
| $P_k \in \Delta^{K-1}$ | Set generative probability for documents or reference information from class $k$. |
| $\mathcal{D}_k \subseteq [V]$ | Set of words specific in class $k$ in the corpus documents. |
| $\mathcal{R}_k \subseteq [V]$ | Set of words specific in class $k$ in the reference information documents |
| $\mathcal{R}_0 \subset [V]$ | Set of words that do not appear in the reference information at all. |
| $\theta_k \in \Delta^{V-1}$ | Word generative probability for corpus documents from class $k$ (unobserved). |
| $r_k \in \Delta^{V-1}$ | Word generative probability for class $k$'s reference information document (observed). |
| $L_n \in \mathbb{N}$ | Length of the $n$th corpus document (random variable). |
| $y_n \in [K]$ | True label for corpus document $n$ (random variable). |
| $X_{nv} \in \mathbb{N}$ | Count of word $v \in \mathcal{V}$ in corpus document $n \in [N]$ (random variable) |

Table 3: Overview of Notation

3. Draw the count of each word $X_{nv}$ with a multinomial distribution with $L_n$ trials and probability given by $\theta_{y_n}$, i.e.

$$(X_{n1}, \ldots, X_{nv}) \mid y_n, L_n \sim \text{Multinomial}(L_n, \theta_{y_n})$$

Alternatively, by the Poisson Thinning property, the distributions of $X_{n1}, \ldots, X_{nv}$ are independent and drawn according to the following distribution

$$X_{nv} \mid y_n = k \sim \text{Pois}(L_0 \cdot \theta_k) \quad \forall v \in [V]$$

The most crucial aspect of this model is the construction of $\{\theta_k\}_{k=1}^K$ and $(\theta_k^{(0)})_{k=1}^K$. We first present the construction, then explain the intuition behind it.

**Constructing $\{\theta_k\}_{k=1}^K$.** The probability distributions $\{\theta_k\}_{k=1}^K$ is computed in a hierarchical fashion from two underlying quantities: (1) a set generating matrix $P \in \mathbb{R}^{K \times K}$ where $P_k \in \Delta^{K-1}$, and (2) a partitioning of $[V]$ into sets $\{\mathcal{D}_k\}_{k=1}^K$. For a word $v$ that belong to the set $\mathcal{D}_k$, $\theta_{kv}$ is given by equation equation 27 below.

$$\theta_{kv} = \frac{1}{|\mathcal{D}_{k'}|} \cdot P_{k,k'} \quad \forall k \in [K], v \in [V], v \in \mathcal{D}_{k'} \tag{27}$$

Equation equation 27 states that the probability of drawing word $v \in \mathcal{D}_{k'}$ from a corpus document with label $y = k$ can be decomposed into two probabilities: First, the set $\mathcal{D}_{k'}$ is chosen with probability $P_{kk'}$, then the word $v$ has a $\frac{1}{|\mathcal{D}_{k'}|}$ chance of being drawn from $\mathcal{D}_{k'}$, i.e. a word is uniformly drawn at random from $\mathcal{D}_{k'}$. In this sense, $\theta_k$ is constructed in a *hierarchical* fashion, where the first-level probabilities are given by $P$ and the second-level are given by the $\mathcal{D}$'s.

**Constructing $(r_k)_{k=1}^K$.** The construction of $r_k$ is similar to that of $\theta_k$. The probability distributions $(r_k)_{k=1}^K$ are computed from two underlying quantities: (1) the same set generating matrix $P \in \mathbb{R}^{X \times X}$ where

$P_k \in \Delta^{K-1}$, and (2) a different partitioning of $[V]$ into the sets $\{\mathcal{R}_k\}_{k=0}^K$. For a word $v$ that belongs to the set $\mathcal{R}_{k'}$, $r_{kv}$ is given by equation equation 28 below.

$$\theta_{kv}^{(0)} = \begin{cases} \frac{1}{|\mathcal{R}_{k'}|} \cdot P_{k,k'} & k' \neq 0 \\ 0 & k' = 0 \end{cases} \quad \forall k \in [K], v \in [V], v \in \mathcal{R}_{k'} \tag{28}$$

A similar interpretation can be applied to equation equation 28: the reference information documents constructs $r_{kv}$ by first choosing a set $\mathcal{R}_{k'}$ that $v$ belongs to according to $P$, then uniformly chooses a word from $\mathcal{R}_{k'}$, thus having a $\frac{1}{|\mathcal{R}_{k'}|}$ chance of choosing $v$. For words $v \in \mathcal{R}_0$, the reference information documents has $r_{kv} = 0$ meaning the reference information document will never generate this word.

A nice property of the Noisy Vocabulary Model is that the construction of $\{\theta_k\}_{k=1}^K$ and $\{r_k\}_{k=1}^K$ have the nice property below:

**Lemma E.1.** *The estimates $(r_k)_{k=1}^K$ of $\{\theta_k\}_{k=1}^K$ preserves the KL divergence between classes, i.e.*

$$KL(\theta_k, \theta_{k'}) = KL(r_k, r_{k'}) \text{ for any } k, k' \in [K] \tag{29}$$

*Proof.* Proof of Lemma E.1 The key property of our model that gives the desired property is that

$$\frac{\theta_{kv}}{\theta_{k',v}} = \frac{P_{k,D(v)}}{P_{k',D(v)}} .$$

for all $v$. The same is true for $r$, namely for any $v$ such that $R(v) \neq 0$, we have

$$\frac{r_{kv}}{r_{k',v}} = \frac{P_{k,D(v)}}{P_{k',D(v)}} .$$

This gives the following result:

$$KL(\theta_k, \theta_{k'}) = \sum_{v \in [V]} \theta_{kv} \cdot \log\left(\frac{\theta_{kv}}{\theta_{k'v}}\right) = \sum_{\ell \in [K]} P_{k\ell} \cdot \log\left(\frac{P_{k\ell}}{P_{k'\ell}}\right)$$

Likewise, we have the same reasoning for $r_k$ and $r_{k'}$

$$KL(r_k, r_k) = \sum_{v \in [V] : r_{kv} \neq 0} r_{kv} \cdot \log\left(\frac{r_{kv}}{r_{k'v}}\right) = \sum_{\ell \in [K]} P_{k\ell} \cdot \log\left(\frac{P_{k\ell}}{P_{k',\ell}}\right)$$

We note that $r_k$ can potentially include 0 entries, but the KL divergence is well defined because whenever $r_{kv} = 0$, then $r_{k',v} = 0$ for all $k' \neq k$ because $r_{kv} = 0$ implies that $v \in \mathcal{R}_0$. □ □

# F   Appendix to Section 6

By the well-known thinning property of Poisson random variables, we can characterize the distributions of the variables $X_{AA}, \ldots, X_{BM}$ conditioned on $Y = A$ or $Y = B$. Define the following proportions of good, corrupt and missing words:

$$p_G = \frac{G}{G + C + M} \qquad p_C = \frac{B}{G + C + M} \qquad p_M = \frac{M}{G + C + M}$$

Then, Table 4 gives the distributions of $X_{AA}, \ldots, X_{B0}$ conditioned on $Y = A$ or $Y = B$. For example, $X_{AA}|(Y = A) \sim \text{Pois}(L_0 \cdot \lambda \cdot p_G)$. Conditioned on $Y = A$, the random variable $X_{AA}$ is the sum of the rate parameters of $G$ possible words, each of which has a $\frac{\lambda}{G+C+M}$ chance of being drawn (recall Figure 10(a)), giving an overall rate parameter of $L_0 \cdot \lambda \cdot p_G$.

**Lemma F.1** (Initial Decision Boundary)**.** *The initial decision boundary for a document with word counts given by $X$ is:*

| | $X_{AA}$ | $X_{AB}$ | $X_{A0}$ |
|---|---|---|---|
| $Y = A$ | $L_0 \cdot \lambda \cdot p_G$ | $L_0 \cdot \lambda \cdot p_C$ | $L_0 \cdot \lambda \cdot p_M$ |
| $Y = B$ | $L_0 \cdot (1 - \lambda) \cdot p_G$ | $L_0 \cdot (1 - \lambda) \cdot p_C$ | $L_0 \cdot (1 - \lambda) \cdot p_M$ |
| | $X_{BB}$ | $X_{BA}$ | $X_{B0}$ |
| $Y = A$ | $L_0 \cdot (1 - \lambda) \cdot p_G$ | $L_0 \cdot (1 - \lambda) \cdot p_C$ | $L_0 \cdot (1 - \lambda) \cdot p_M$ |
| $Y = B$ | $L_0 \cdot \lambda \cdot p_G$ | $L_0 \cdot \lambda \cdot p_C$ | $L_0 \cdot \lambda \cdot p_M$ |

Table 4: Rate Parameters of $X_{AA}, \ldots, X_{B0}$ conditioned on $Y$

$$X_{AA} - X_{BB} - (X_{AB} - X_{BA}) > 0 \quad \Leftrightarrow \quad Y^{(0)} = A \tag{30}$$
$$X_{AA} - X_{BB} - (X_{AB} - X_{BA}) < 0 \quad \Leftrightarrow \quad Y^{(0)} = B \tag{31}$$

We ignore the case when the above is equal to 0. Those documents can just be randomly assigned to $A$ or $B$.

*Proof.* Proof Equation equation 3 in the DCRI with MNB Methodology gives the following decision boundary for the initial label being $A$:

$$\sum_{v \in [V]} X_v \cdot [\log r_{Av} - \log r_{Bv}] > 0 \quad \Leftrightarrow \quad Y^{(0)} = A \tag{32}$$

By the definitions of $r_A$ and $r_B$ from Figure 10(a), $\log r_{Av} - \log r_{Bv}$ can be divided into cases:

$$\log r_{Av} - \log r_{Bv} = \begin{cases} \log\left(\frac{\lambda}{1-\lambda}\right) & v \in \mathcal{V}_{AA} \\ -\log\left(\frac{\lambda}{1-\lambda}\right) & v \in \mathcal{V}_{BB} \\ -\log\left(\frac{\lambda}{1-\lambda}\right) & v \in \mathcal{V}_{AB} \\ \log\left(\frac{\lambda}{1-\lambda}\right) & v \in \mathcal{V}_{BA} \end{cases}$$

We ignore the words $v \in \mathcal{V}_{A,M} \cup \mathcal{V}_{B,M}$ since $r_{Av} = r_{Bv} = 0$ for such words. Dividing through by $\log\left(\frac{\lambda}{1-\lambda}\right)$ gives equation equation 30. A similar derivation applies to equation equation 31. $\square$

**Lemma F.2** (Iterated Decision Boundary). *Given $Y^{(0)}$, the updated decision boundary for $Y^{(1)}$ is given by:*

$$g \cdot (X_{AA} - X_{BB}) + b \cdot (X_{AB} - X_{BA}) + m \cdot (X_{A0} - X_{B0}) > 0 \quad \Leftrightarrow \quad Y^{(1)} = A \tag{33}$$

*where $g := \log \frac{E[X_{AA}|Y^{(0)}=A]}{E[X_{AA}|Y^{(0)}=B]}$, $b := \log \frac{E[X_{AB}|Y^{(0)}=A]}{E[X_{AB}|Y^{(0)}=B]}$ and $m := \log \frac{E[X_{A0}|Y^{(0)}=A]}{E[X_{A0}|Y^{(0)}=B]}$.*

*Proof.* Proof Let $\Theta_A^{(1)}$ and $\Theta_B^{(1)}$ be random vectors for the distributions estimated from the documents with initial label $Y^{(0)} = A$ and $Y^{(0)} = B$ respectively. From Equation equation 5 of the DCRI with MNB methodology, we have that:

$$\Theta_{kv}^{(1)} = \frac{\sum_{n=1}^{N} \mathbb{I}_{Y_n^{(0)}=k} \cdot X_{nv}}{\sum_{n=1}^{N} \mathbb{I}_{Y_n^{(0)}=k} \cdot L_n} \qquad k \in \{A, B\} \tag{34}$$

As we take the limit of $N \to \infty$, the random vectors $\Theta_A^{(1)}$ and $\Theta_B^{(1)}$ will converge to constant vectors $\theta_A^{(1)}, \theta_B^{(1)}$. This gives us the following result.

$$
\begin{aligned}
\theta_{Av}^{(1)} = \lim_{N\to\infty} \Theta_{Av}^{(1)} &= \lim_{N\to\infty} \frac{\sum_{n=1}^{N} \mathbb{I}\{Y_n^{(0)} = A\} \cdot X_{nv}}{\sum_{n=1}^{N} \mathbb{I}\{Y_n^{(0)} = A\} \cdot L_n} = \lim_{N\to\infty} \frac{\frac{1}{N} \cdot \sum_{n=1}^{N} \mathbb{I}\{Y_n^{(0)} = A\} \cdot X_{nv}}{\frac{1}{N} \cdot \sum_{n=1}^{N} \mathbb{I}\{Y_n^{(0)} = A\} \cdot L_n} \\
&= \frac{E[X_v \cdot \mathbb{I}\{Y^{(0)} = A\}]}{E[L_n \cdot \mathbb{I}\{Y^{(0)} = A\}]} = \frac{E[X_v|Y^{(0)} = A] \cdot P(Y^{(0)} = A)}{L_0 \cdot P(Y^{(0)} = A)} \\
&= \frac{E[X_v|Y^{(0)} = A]}{L_0} \\
\theta_{Bv}^{(1)} = \lim_{N\to\infty} \Theta_{Bv}^{(1)} &= \frac{E[X_v|Y^{(0)} = B]}{L_0} \\
\log \theta_{Av}^{(1)} - \log \theta_{Bv}^{(1)} &= \log \frac{E[X_v|Y^{(0)} = A]}{E[X_v|Y^{(0)} = B]}
\end{aligned}
$$

The derivation for $\theta_{Av}^{(1)}$ above used the law of large numbers to convert empirical means into expectations. The decision boundary can be constructed as follows:

$$\sum_{v\in[V]} X_v \cdot \log \frac{E[X_v|Y^{(0)} = A]}{E[X_v|Y^{(0)} = B]} > 0 \quad \Leftrightarrow \quad Y^{(1)} = A \tag{35}$$

To get this to equation equation 33, we apply the following two observations:

**Observation 1.** All words $v \in \mathcal{V}_{AA}$ have the same value of $\log \frac{E[X_v|Y^{(0)}=A]}{E[X_v|Y^{(0)}=B]}$. The same is true for the sets $\mathcal{V}_{AB}, \dots, \mathcal{V}_{B,M}$. For example:

$$E[X_v|Y^{(0)} = A] = E\left[E[X_v|Y^{(0)} = A, X_{AA}] \mid Y^{(0)} = A\right] = \frac{1}{G} \cdot E[X_{AA} \mid Y^{(0)} = A] \tag{36}$$

$$E[X_v|Y^{(0)} = B] = E\left[E[X_v|Y^{(0)} = B, X_{AA}] \mid Y^{(0)} = A\right] = \frac{1}{G} \cdot E[X_{AA} \mid Y^{(0)} = B] \tag{37}$$

This implies the fact that the coefficients of $X_v$ for $v \in \mathcal{V}_{AA}$ in equation equation 35 are all the same.

$$\log \frac{E[X_v|Y^{(0)} = A]}{E[X_v|Y^{(0)} = B]} = \log \frac{E[X_{AA}|Y^{(0)} = A]}{E[X_{AA}|Y^{(0)} = B]} \quad \forall v \in \mathcal{V}_{AA}$$

A similar argument can be applied to $\mathcal{V}_{AB}, \dots, \mathcal{V}_{B,M}$. This results in the following decision boundary:

$$X_{AA} \cdot \log \frac{E[X_{AA}|Y^{(0)} = A]}{E[X_{AA}|Y^{(0)} = B]} + \dots + X_{B0} \cdot \log \frac{E[X_{B0}|Y^{(0)} = A]}{E[X_{B0}|Y^{(0)} = B]} > 0 \quad \Leftrightarrow \quad Y^{(1)} = A \tag{38}$$

**Observation 2.** Symmetry between $\mathcal{V}_{AA}$ and $\mathcal{BB}$ implies that $g := \log \frac{E[X_{AA}|Y^{(0)}=A]}{E[X_{AA}|Y^{(0)}=B]} = -\log \frac{E[X_{BB}|Y^{(0)}=A]}{E[X_{BB}|Y^{(0)}=B]}$. A word $v \in \mathcal{V}_{AA}$ is equivalent to a word in $\mathcal{V}_{BB}$ if the labels $A$ and $B$ are flipped. This gives:

$$\log \frac{E[X_{AA}|Y^{(0)} = A]}{E[X_{AA}|Y^{(0)} = B]} = \log \frac{E[X_{BB}|Y^{(0)} = B]}{E[X_{BB}|Y^{(0)} = A]} = -\log \frac{E[X_{BB}|Y^{(0)} = A]}{E[X_{BB}|Y^{(0)} = B]}$$

The same is true for the pairs $(\mathcal{V}_{AB}, \mathcal{V}_{BA})$ and $(\mathcal{V}_{A,M}, \mathcal{V}_{B,M})$. Therefore, out of the 6 terms in equation equation 38, we can pair them up as $g, -g, b, -b, m, -m$, which provides the desired equation equation 33. Dividing through by $g$ gives the format of the decision boundary shown in Figure 10(b), with $\alpha = \frac{b}{g}$ and $\beta = \frac{m}{g}$. $\qquad\square$ $\qquad\square$

Our main claim regarding this decision boundary is that $\frac{b}{g} \geq -1 (i.e., \alpha \geq -1$ and $\frac{m}{g} \geq 0$ (i.e., $\beta \geq 0$), claimed in Lemmas F.4 and F.5 respectively. The following Lemma F.3 proves a short claim regarding just $g$.

**Lemma F.3** ($g$ is Positive). $g > 0$

*Proof.* Proof We must show that $E[X_{AA}|Y^{(0)} = A] > E[X_{AA}|Y^{(0)} = B]$. We break this inequality into two pieces:

$$
\begin{aligned}
E[X_{AA}|Y^{(0)} = A] &= E[X_{AA}|Y^{(0)} = A, Y = A] \cdot \rho + E[X_{AA}|Y^{(0)} = A, Y = B] \cdot (1 - \rho) \\
&> E[X_{AA}|Y = A] \cdot \rho + E[X_{AA}|Y = B] \cdot (1 - \rho) \\
&= L_0 \cdot p_G \cdot [\lambda \cdot \rho + (1 - \lambda)(1 - \rho)] \\
E[X_{AA}|Y^{(0)} = B] &= E[X_{AA}|Y^{(0)} = B, Y = A] \cdot (1 - \rho) + E[X_{AA}|Y^{(0)} = B, Y = B] \cdot \rho \\
&< E[X_{AA}|Y = A] \cdot (1 - \rho) + E[X_{AA}|Y = B] \cdot \rho \\
&= L_0 \cdot p_G \cdot [\lambda \cdot (1 - \rho) + (1 - \lambda)\rho]
\end{aligned}
$$

In inequalities above stem from the fact that for $k \in \{A, B\}$

$$
\begin{aligned}
E[X_{AA}|Y^{(0)} = A, Y = k] &= E[X_{AA}|X_{AA} \underbrace{-X_{BB} - X_{AB} + X_{BB}}_{-Z} > 0, Y = k] \\
&= E[X_{AA}|X_{AA} > Z, Y = k] \\
&> E[X_{AA}|Y = k] = L_0 \cdot p_G \cdot \lambda
\end{aligned}
$$

The conditional expectation of $X_{AA}$ given the event $X_{AA} > Z$ is higher than the marginal expectation of $X_{AA}$. To see this, consider any value $z$ that $Z$ takes on. Regardless of whether $z$ is positive or negative, $E[X_{AA} \mid X_{AA} > z, Y^{(0)} = A] \geq E[X_{AA} \mid Y^{(0)} = A]$. The $\geq$ holds with strict $>$ for any $z \geq 0$ and thus a strict inequality holds overall. By a similar reasoning, we get that $E[X_{AA} \mid Y^{(0)} = B, Y = k] < E[X_{AA} \mid Y = k]$ for $k \in \{A, B\}$. $\qquad\square$ $\qquad\square$

**Lemma F.4** (Bad Words Reduced). $\frac{b}{g} \geq -1$ *(i.e., $\alpha \geq -1$)*

*Proof.* Proof

$\frac{b}{g} \geq -1$ equivalent to $b \geq -g$ since $g > 0$. This claim is equivalent to:

$$
\frac{E[X_{AB}|Y^{(0)} = A]}{E[X_{AB}|Y^{(0)} = B]} \geq \frac{E[X_{AA}|Y^{(0)} = B]}{E[X_{AA}|Y^{(0)} = A]}
$$

Using symmetry of class $A$ and $B$ and the fact that all terms above are positive, we can show the following equivalent inequality:

$$
\frac{E[X_{AB}|Y^{(0)} = A]}{E[X_{BB}|Y^{(0)} = A]} \geq \frac{E[X_{BA}|Y^{(0)} = A]}{E[X_{AA}|Y^{(0)} = A]} \tag{39}
$$

Expand out the terms by conditioning on $Y = A$ or $Y = B$ and define the terms $x, x', y, y', z, z', w$ and $w'$ for notational convenience as follows:

$$\frac{E[X_{AB}|Y^{(0)} = A]}{E[X_{BB}|Y^{(0)} = A]} = \frac{\rho \cdot \overbrace{E[X_{AB}|Y = A, Y^{(0)} = A]}^{x} + (1-\rho) \cdot \overbrace{E[X_{AB}|Y = B, Y^{(0)} = A]}^{y}}{\rho \cdot \underbrace{E[X_{BB}|Y = A, Y^{(0)} = A]}_{x'} + (1-\rho) \cdot \underbrace{E[X_{BB}|Y = B, Y^{(0)} = A]}_{y'}} \tag{40}$$

$$\frac{E[X_{BA}|Y^{(0)} = A]}{E[X_{AA}|Y^{(0)} = A]} = \frac{\rho \cdot \overbrace{E[X_{BA}|Y = A, Y^{(0)} = A]}^{z} + (1-\rho) \cdot \overbrace{E[X_{BA}|Y = B, Y^{(0)} = A]}^{w}}{\rho \cdot \underbrace{E[X_{AA}|Y = A, Y^{(0)} = A]}_{z'} + (1-\rho) \cdot \underbrace{E[X_{AA}|Y = B, Y^{(0)} = A]}_{w'}} \tag{41}$$

We make the following two claims

**Claim 1. Relationship between** $(x, x')$, $(y, y'), (z, z')$ **and** $(w, w')$

$$\frac{x}{x'} = \frac{E[X_{AB}|Y = A, Y^{(0)} = A]}{E[X_{BB}|Y = A, Y^{(0)} = A]}$$
$$= \frac{E[E[X_{AB}|Y = A, Y^{(0)} = A, X_{AB} + X_{BB}] \mid Y = A, Y^{(0)} = A]}{E[E[X_{BB}|Y = A, Y^{(0)} = A, X_{AB} + X_{BB}] \mid Y = A, Y^{(0)} = A]}$$

Conditioning on $X_{AB} + X_{BB}$ makes $X_{AB}$ is now independent of $Y^{(0)} = A$. To see this, recall that $Y^{(0)} = A$ is equivalent to $X_{AA} + X_{BA} > X_{AB} + X_{BB}$. Taking the right-hand-side as a constant, the condition $X_{AA} + X_{BA} > z$ is independent of $X_{BB}$ and $X_{AB}$.

$$= \frac{E[E[X_{AB}|Y = A, X_{AB} + X_{BB}] \mid Y = A, Y^{(0)} = A]}{E[E[X_{BB}|Y = A, X_{AB} + X_{BB}] \mid Y = A, Y^{(0)} = A]}$$
$$= \frac{E\left[\frac{\lambda \cdot p_C}{\lambda \cdot p_C + (1-\lambda) \cdot p_G} \cdot (X_{AB} + X_{BB}) \mid Y = A, Y^{(0)} = A\right]}{E\left[\frac{(1-\lambda) \cdot p_G}{\lambda \cdot p_C + (1-\lambda) \cdot p_G} \cdot (X_{AB} + X_{BB}) \mid Y = A, Y^{(0)} = A\right]} = \frac{\lambda \cdot p_C}{(1 - \lambda) \cdot p_G}$$

In the last equality, we used the fact that $X_{AB}$ and $X_{BB}$ are independent Poisson random variables. By the Poisson thinning property, conditioned on the sum $X_{AB} + X_{BB}$, we know that $X_{AB}$ is a binomial random variable over $X_{AB} + X_{BB}$ trials and probability of success $\frac{E[X_{AB}|Y=A]}{E[X_{AB}|Y=A]+E[X_{BB}|Y=A]} = \frac{\lambda \cdot p_C}{\lambda \cdot p_C + (1-\lambda) \cdot p_G}$. A similar calculation is done for $E[X_{BB} \mid Y = A]$.

By an analogous calculation, we obtain the following 4 equalities:

$$\frac{x}{x'} = \frac{\lambda \cdot p_C}{(1 - \lambda) \cdot p_G} \qquad\qquad \frac{y}{y'} = \frac{(1 - \lambda) \cdot p_C}{\lambda \cdot p_G}$$
$$\frac{z}{z'} = \frac{(1 - \lambda) \cdot p_C}{\lambda \cdot p_G} \qquad\qquad \frac{w}{w'} = \frac{\lambda \cdot p_C}{(1 - \lambda) \cdot p_G}$$

**Claim 2. Relationship between** $(x + x', y + y')$ **and** $(z + z', w + w')$. We claim the following

$$\rho \cdot (x + x') \geq (1 - \rho) \cdot (y + y')$$
$$\rho \cdot (z + z') \geq (1 - \rho) \cdot (w + w')$$

Consider the first inequality involving $x, x', y, y'$. We can write it out as follows:

$$\frac{\rho \cdot (x + x')}{(1 - \rho) \cdot (y + y')} = \frac{E[(X_{AB} + X_{BB}) \cdot I\{X_{AB} + X_{BB} < X_{AA} + X_{BA}\} \mid Y = A]}{E[(X_{AB} + X_{BB}) \cdot I\{X_{AB} + X_{BB} < X_{AA} + X_{BA}\} \mid Y = B]} \geq 1$$

For notational convenience, let $Z = X_{AB} + X_{BB}$ and $W = X_{AA} + X_{BA}$. Consider any $(z, w) \in \mathbb{N}$ such that $z < w$. We claim the following:

$$\frac{P(Z = z, W = w \mid Y = A)}{P(Z = z, W = w \mid Y = B)} = \frac{[L_0 \cdot (p_C \cdot \lambda + p_G \cdot (1 - \lambda))]^z \cdot [L_0 \cdot (p_G \cdot \lambda + p_C \cdot (1 - \lambda))]^w}{[L_0 \cdot (p_G \cdot \lambda + p_C \cdot (1 - \lambda))]^z \cdot [L_0 \cdot (p_C \cdot \lambda + p_G \cdot (1 - \lambda))]^w}$$

$$= \left( \frac{p_C \cdot \lambda + p_G \cdot (1 - \lambda)}{p_G \cdot \lambda + p_C \cdot (1 - \lambda)} \right)^{z - w}$$

$$\geq 1$$

The first equality is true by recalling that $Z$ and $W$ are just independent Poisson random variables conditioned on $Y$. Their rate parameters can be found in Table 4. The second equality uses the PMF of the Poisson Distribution, while the third inequality uses the fact that we assumed $z - w < 0$ and that $p_G \cdot \lambda + p_G \cdot (1 - \lambda) < p_G \cdot \lambda + p_C \cdot (1 - \lambda)$ given that $\lambda > \frac{1}{2}$ and $p_G > p_C$ (more good words than corrupted words).

A similar argument can be used to show that $\frac{\rho(z + z')}{(1 - \rho) \cdot (w + w')} \geq 1$. Note that $x$ and $z$ condition on the same $Y = A, Y^{(0)} = A$ and only differ in what the expectation is taken on, which is not used in our argument above. We only used the fact that $(X_{AB} + X_{BB})$ is the same in the numerator and denominator of $\frac{\rho \cdot (x + x')}{(1 - \rho) \cdot (y + y')}$.

**Putting it Together.** Finally, we put together Claim 1 and 2 to obtain the desired expression through the following inequalities:

$$\frac{E[X_{AB} | Y^{(0)} = A]}{E[X_{BB} | Y^{(0)} = A]} = \frac{\rho \cdot x + (1 - \rho) \cdot y}{\rho \cdot x' + (1 - \rho) \cdot y'} \geq \frac{1}{2} \cdot \frac{p_G}{p_C} \left( \frac{\lambda}{1 - \lambda} + \frac{1 - \lambda}{\lambda} \right)$$

$$\frac{E[X_{BA} | Y^{(0)} = A]}{E[X_{AA} | Y^{(0)} = A]} = \frac{\rho \cdot z + (1 - \rho) \cdot w}{\rho \cdot z' + (1 - \rho) \cdot w'} \leq \frac{1}{2} \cdot \frac{p_G}{p_C} \left( \frac{1 - \lambda}{\lambda} + \frac{\lambda}{1 - \lambda} \right)$$

Consider the expression:

$$\frac{\rho \cdot x + (1 - \rho) \cdot y}{\rho \cdot x' + (1 - \rho) \cdot y'}$$

It is clear that a lower bound on this expression is obtained when $\rho \cdot (x + x') = (1 - \rho)(y + y')$. This expression is a weighted ratio of $\frac{x}{x'}$ and $\frac{y}{y'}$ where recall from Claim 1 that $\frac{x}{x'} \geq \frac{y}{y'}$. The weighting is based on the relative sizes of $\rho \cdot x$ and $(1 - \rho) \cdot y$. Therefore, given the constraint $\rho(x + x') \geq (1 - \rho)(y + y')$, a lower bound is obtained at equality. A similar argument can be made for the term $\frac{\rho \cdot z + (1 - \rho) \cdot w}{\rho \cdot z' + (1 - \rho) \cdot w'}$.

This leads to:

$$\frac{\rho \cdot x + (1 - \rho) \cdot y}{\rho \cdot x' + (1 - \rho) \cdot y'} = \frac{\frac{\rho \cdot x + (1 - \rho) \cdot y}{\rho \cdot (x + x') + (1 - \rho) \cdot (y + y')}}{1 - \frac{\rho \cdot x + (1 - \rho) \cdot y}{\rho \cdot (x + x') + (1 - \rho) \cdot (y + y')}}$$

$$\geq \frac{\frac{\rho \cdot x}{\rho(x + x')} + \frac{(1 - \rho) \cdot y}{(1 - \rho) \cdot (y + y')}}{1 - \frac{\rho \cdot x}{\rho(x + x')} - \frac{(1 - \rho) \cdot y}{(1 - \rho) \cdot (y + y')}}$$

$$= \frac{\frac{1}{1 + \frac{x'}{x}} + \frac{1}{1 + \frac{y'}{y}}}{1 - \frac{1}{1 + \frac{x'}{x}} - \frac{1}{1 + \frac{y'}{y}}}$$

$$\frac{\rho \cdot z + (1 - \rho) \cdot w}{\rho \cdot z' + (1 - \rho) \cdot w'} = \frac{\frac{\rho \cdot z + (1-\rho) \cdot w}{\rho \cdot (z+z') + (1-\rho) \cdot (w+w')}}{1 - \frac{\rho \cdot z + (1-\rho) \cdot w}{\rho \cdot (z+z') + (1-\rho) \cdot (w+w')}}$$

$$\leq \frac{\frac{\rho \cdot z}{\rho(z+z')} + \frac{(1-\rho) \cdot w}{(1-\rho) \cdot (w+w')}}{1 - \frac{\rho \cdot z}{\rho(z+z')} - \frac{(1-\rho) \cdot w}{(1-\rho) \cdot (w+w')}}$$

$$= \frac{\frac{1}{1 + \frac{z'}{z}} + \frac{1}{1 + \frac{w'}{w}}}{1 - \frac{1}{1 + \frac{z'}{z}} - \frac{1}{1 + \frac{w'}{w}}}$$

Since $\frac{z'}{z} = \frac{y'}{y}$ and $\frac{x'}{x} = \frac{w'}{w}$, this gives the desired result. This concludes the proof. □

□

**Lemma F.5** (Missing Words Amplified). $\frac{m}{g} \geq 0$ *(i.e., $\beta \geq 0$)*

*Proof.* Proof of Lemma F.5 Lemma F.4 already showed that $g > 0$, so it is sufficient to show that $m \geq 0$. This is equivalent to showing:

$$\frac{E[X_{A0} \mid Y^{(0)} = A]}{E[X_{A0} \mid Y^{(0)} = B]} \geq 1$$

Conditioning on $Y = A$ and $Y = B$ gives:

$$E[X_{A0} \mid Y^{(0)} = A] = \rho \cdot E[X_{A0} \mid Y^{(0)} = A, Y = A] + (1 - \rho) \cdot E[X_{A0} \mid Y^{(0)} = A, Y = B]$$
$$= \rho \cdot L_0 \cdot \lambda \cdot p_M + (1 - \rho) \cdot L_0 \cdot (1 - \lambda) \cdot p_M$$
$$E[X_{A0} \mid Y^{(0)} = B] = (1 - \rho) \cdot E[X_{A0} \mid Y^{(0)} = B, Y = A] + \rho cdot E[X_{A0} \mid Y^{(0)} = B, Y = B]$$
$$= (1 - \rho) \cdot L_0 \cdot \lambda \cdot p_M + \rho \cdot L_0 \cdot (1 - \lambda) \cdot p_M$$

The events $Y^{(0)} = A$ or $Y^{(0)} = B$ are independent of $X_{A0}$. This gives the desired inequality:

$$\lambda \cdot \rho + (1 - \lambda) \cdot (1 - \rho) \geq (1 - \rho) \cdot \lambda + \rho \cdot (1 - \lambda)$$

which is true given $\rho, \lambda > \frac{1}{2}$. This concludes the proof □

□

**Lemma F.6** (Accuracy Increases). *Without loss of generality, assume that $Y = A$. The probability that $Y^{(1)} = A$ is at least as large as the probability that $Y^{(0)} = A$, i.e.*

$$P(Y^{(1)} = A \mid Y = A) \geq P(Y^{(0)} = A \mid Y = A)$$

*Recall from Lemma F.1 and Lemma F.2 that the decision boundaries for $Y^{(1)} = A$ and $Y^{(0)} = A$ are as follows:*

$$(X_{AA} - X_{BB}) + \alpha \cdot \cdot (X_{AB} - X_{BA}) + \beta \cdot (X_{A0} - X_{B0}) > 0 \quad \Leftrightarrow \quad Y^{(1)} = A$$
$$(X_{AA} - X_{BB}) - 1 \cdot (X_{AB} - X_{BA}) + 0 \cdot (X_{A0} - X_{B0}) > 0 \quad \Leftrightarrow \quad Y^{(0)} = A$$

*where $\frac{b}{g} \geq -1$ (Lemma F.4) and $\frac{m}{g} \geq 0$ (Lemma F.5)*

*Proof.* Proof of Lemma F.6

Consider an intermediate decision boundary:

$$(X_{AA} - X_{BB}) + \frac{b}{g} \cdot (X_{AB} - X_{BA}) \quad \Leftrightarrow \quad Y^{(0.5)} = A$$

We will show that

$$P(Y^{(1)} = A \mid Y = A) \geq P(Y^{(0.5)} = A \mid Y = A) \geq P(Y^{(0)} = A \mid Y = A)$$

**Step 1.** First, $P(Y^{(0.5)} = A \mid Y = A) \geq P(Y^{(0)} = A \mid Y = A)$. An equivalent inequality is:

$$P((X_{AA}, X_{BB}, X_{AB}, A_{BA}) \in \mathcal{G}) \geq P((X_{AA}, X_{BB}, X_{AB}, A_{BA}) \in \mathcal{B}) \tag{42}$$

where the sets $\mathcal{G}$ and $\mathcal{B}$ (representing good and bad) are defined below.

$$\mathcal{G} = \left\{ (x, y, z, w) \in \mathbb{N}^4 \mid x - y + \frac{b}{g} \cdot (z - w) > 0 \text{ and } x - y - 1 \cdot (z - w) < 0 \right\}$$

$$\mathcal{B} = \left\{ (x, y, z, w) \in \mathbb{N}^4 \mid x - y + \frac{b}{g} \cdot (z - w) < 0 \text{ and } x - y - 1 \cdot (z - w) > 0 \right\}$$

To show this, note that there is a $1 : 1$ mapping between $\mathcal{G}$ and $\mathcal{B}$ by swapping the values of $(x, y)$ and $(z, w)$. We claim the following:

For *any* $(x, y, z, w) \in \mathcal{G}$,

$$P((X_{AA}, X_{BB}, X_{AB}, X_{BA}) = (x, y, z, w)) \geq P((X_{AA}, X_{BB}, X_{AB}, X_{BA}) = (y, x, w, z))$$

This is true because:

$$\frac{P((X_{AA}, X_{BB}, X_{AB}, X_{BA}) = (x, y, z, w))}{P((X_{AA}, X_{BB}, X_{AB}, X_{BA}) = (y, x, w, z))} = \frac{(L_0 \cdot p_G \cdot \lambda)^x \cdot (L_0 \cdot p_G \cdot (1 - \lambda))^y \cdot (L_0 \cdot p_C \cdot \lambda)^z \cdot (L_0 \cdot p_C \cdot (1 - \lambda))^w}{(L_0 \cdot p_G \cdot \lambda)^y \cdot (L_0 \cdot p_G \cdot (1 - \lambda))^x \cdot (L_0 \cdot p_C \cdot \lambda)^w \cdot (L_0 \cdot p_C \cdot (1 - \lambda))^z}$$

$$= \left( \frac{\lambda}{1 - \lambda} \right)^{x - y + z - w}$$

From the definition of $\mathcal{G}$ and the fact that $\frac{b}{g} \geq -1$, we see that the only way for $x - y + \frac{b}{g} \cdot (z - w) > 0$, but $x - y - (z - w) < 0$ is for $z - w > 0$. Thus, $x + y + z - w > x + y + \frac{b}{g} \cdot (z - w) > 0$ and since $\left( \frac{\lambda}{1 - \lambda} \right) > 1$, we get the desired result.

**Step 2.** Next, $P(Y^{(1)} = A \mid Y = A) \geq P(Y^{(0.5)} = A \mid Y = A)$. Similar to before, define the following sets $\mathcal{G}$ and $\mathcal{B}$, which have a $1 : 1$ correspondence:

$$\mathcal{G} = \left\{ (x, y, z, w, u, v) \in \mathbb{N}^6 \mid x - y + \frac{b}{g} \cdot (z - w) < 0 \text{ and } x - y + \frac{b}{g} \cdot (z - w) + \frac{m}{g} \cdot (u - v) > 0 \right\}$$

$$\mathcal{B} = \left\{ (x, y, z, w, u, v) \in \mathbb{N}^6 \mid x - y + \frac{b}{g} \cdot (z - w) > 0 \text{ and } x - y + \frac{b}{g} \cdot (z - w) + \frac{m}{g} \cdot (u - v) < 0 \right\}$$

Similar to before, we have:

$$\frac{P((X_{AA}, X_{BB}, X_{AB}, X_{BA}, X_{A0}, X_{B0}) = (x, y, z, w, u, v))}{P((X_{AA}, X_{BB}, X_{AB}, X_{BA}, X_{A0}, X_{B0}) = (y, x, w, z, v, u))} = \left( \frac{\lambda}{1 - \lambda} \right)^{x - y + z - w + u - v}$$

By a similar argument as before, we have that $(u - v) > 0$ and $(z - w) > 0$ must be true for elements in $\mathcal{G}$. This gives the desired result: $x - y + z - w + u - v > x - y + \frac{b}{g} \cdot (z - w) + \frac{m}{g} \cdot (u - v) > 0$ and thus the ratio above is at least 1. $\square$ $\square$

