# OpenReview forum: "Document Classification using Reference Information"
_TMLR — Rejected by TMLR_

### Review · Reviewer_RS7L · 2025-04-04

**Summary Of Contributions:**

The paper introduces DCRI, which leverages external reference documents to generate initial weak labels for large unlabeled text corpora and then iteratively refines these labels using a Multinomial Naive Bayes classifier, optionally incorporating limited expert feedback. This method efficiently bridges the gap between unsupervised and supervised learning, and reduces the need for extensive manual labeling while achieving near-supervised accuracy, as demonstrated through extensive experiments on pharmaceutical and news datasets and supported by a theoretical analysis based on a generative model.

**Audience:**

Yes

**Broader Impact Concerns:**

-

**Claims And Evidence:**

Yes

**Requested Changes:**

While no mandatory changes are required, a deeper dive into comparisons and analysis regarding the embeddings could further strengthen the impact of the paper.

**Strengths And Weaknesses:**

Strengths:
- the paper is well-written and easy to follow, the presentation is assisted with appropriate illustrative figures.
- motivations are clear, with motivating examples
- the method is novel
- evaluation is comprehensive

Weakness:
- Limited Exploration of Embeddings: The paper primarily relies on a Bag-of-Words representation, which may not capture deeper semantic relationships as effectively as modern LLM embedding-based methods. A comparison with these approaches could provide additional insights.
- Dependence on Reference Documents: The framework assumes that high-quality reference documents exist for each class, which may not always be the case. The method's generality in scenarios where reference documents are noisy or incomplete might be a challenge.
- Generalization to Deep Learning-Based Approaches: While the iterative refinement process improves classification performance, the study does not explore how the approach could integrate with more advanced deep learning models, such as fine-tuned LLMs, which could further enhance performance and is the SoTA approach.

---

> ### Author Response · Authors · 2025-06-03
> **Response to Reviewer**
>
> Thank you for the thorough review of our paper--we sincerely appreciate the feedback. We acknowledge the weaknesses you mentioned and will add them to our paper as limitation of our work. We respond to a few of them below:
>
> Regarding the use of Bag of Words, we’d like to highlight that this choice was made for interpretability and modeling purposes. Our theoretical model (the Noisy Vocabulary model) is based on real-world insights we obtained from analyzing our Bag of Word embeddings, which wouldn’t have been possible with a less interpretable embedding model.
>
> Regarding the limited exploration of embeddings and generalization to deep learning based approaches, we believe that such methods would certainly outperform MNB in the iterative re-training process. Again, for interpretability reasons, MNB allowed us to gather insights as to why this re-training process is helpful (i.e., how the distribution of each word changes), which wouldn’t have been possible with an uninterpretable approach like neural networks.

---

### Review · Reviewer_Kmjd · 2025-05-05

**Summary Of Contributions:**

The paper presents an iterative nearest neighborhood algorithm for document classification that aims to operate in limited annotation settings. The framework leverages reference document information and proposes an iterative approach to classification. The authors attempt to address the challenge of document classification in scenarios where abundant documents are available but annotations are scarce.

**Audience:**

No

**Broader Impact Concerns:**

The paper addresses a relevant problem in document classification, but its current form falls short of demonstrating significant advancement over existing methods. The work would benefit from stronger literature review and more comprehensive experimental validation.

**Claims And Evidence:**

No

**Requested Changes:**

1. Comprehensively update the related work section to include recent advances in:
- Semi-supervised document classification
- Zero-shot learning approaches
- Deep learning-based methods
Discuss how the proposed method relates to or differs from these approaches.

2. Experimental Validation
- Include comparisons with state-of-the-art methods
- Consider adapting and comparing with zero-shot learning approaches
- Evaluate the method on larger-scale datasets to support the motivation

3. Technical Innovation
- Clearly articulate the novel aspects of the methodology
- Strengthen the technical contributions beyond standard components

4. Method Enhancement
- Consider incorporating modern deep learning components
- Explore integration with pre-trained language models

**Strengths And Weaknesses:**

**Strengths**
- Addresses a practical challenge in document classification
- The paper is well-structured and thoroughly presented
- The methodology is clearly explained
- The experimental setup is well-documented

**Weaknesses**
1. Methodological Limitations.
The proposed methodology demonstrates limited innovation:
- Core components rely on well-established techniques (e.g., BoW vectorization).
- Individual phases of the algorithm employ standard approaches.
- The integration of components, while systematic, offers minimal technical novelty.
2. Experimental Design Inconsistencies.
There is a misalignment between the paper's motivation and experimental validation. The authors motivate the work by citing scenarios with "very high" numbers of available documents. However, the experimental validation only considers datasets with up to 6,000 samples
This limitation raises questions about the method's scalability and real-world applicability
3. Outdated Literature Review and Missing Comparisons.
The related work section fails to reflect the current state of the field, particularly:
- No coverage of recent deep learning approaches
- Missing discussion of contemporary semi-supervised and unsupervised methods
- Absence of zero-shot learning approaches

**Example** omissions that are recent and significantly relevant:

Ozmen, M., Cotnareanu, J., Coates, M. (2023). Substituting Data Annotation with Balanced Neighbourhoods and Collective Loss in Multi-label Text Classification. In Proceedings of The 2nd Conference on Lifelong Learning Agents, in Proceedings of Machine Learning Research.

Wang, Y. S., Chi, T. C., Zhang, R., & Yang, Y. (2023). PESCO: Prompt-enhanced Self Contrastive Learning for Zero-shot Text Classification. In Proceedings of the 61st Annual Meeting of the Association for Computational Linguistics (pp. 1234-1245). Association for Computational Linguistics.

Zhang, X., Lewis, M., & Zettlemoyer, L. (2023). Large Language Models are Zero-Shot Document Classifiers. In Transactions of the Association for Computational Linguistics, 11, 914-929.

Meng, Y., Zhang, Y., Huang, J., Wang, X., Zhang, Y., Yu, H., & Han, J. (2023). Text Classification Using Label Names Only: A Language Model Self-Training Approach. In Proceedings of the 2023 Conference on Empirical Methods in Natural Language Processing (EMNLP 2023).

Li, S., Yang, Y., Zhou, C., He, X., & Sun, M. (2023). ContrastNet: A Contrastive Learning Framework for Few-Shot Text Classification. In Findings of the Association for Computational Linguistics: ACL 2023 (pp. 891-903).

4. Absence of comparisons with state-of-the-art methods.
- No adaptation or comparison with adaptable deep learning approaches

---

> ### Author Response · Authors · 2025-06-03
> **Response to Reviewer**
>
> Thank you for the thorough review of our paper--we sincerely appreciate the feedback. We would like to address your concerns regarding the technical innovation of our paper, including comparisons to state-of-the-art methods.
>
> First, we would like to highlight the main contribution of our paper: DCRI is an algorithmic scheme that is modular in that one could use different methods in each of the respective components. Indeed, we describe a scenario with rather naïve methods such as bag of words and MNB, and it is reasonable to hypothesize that more advanced methods will lead to superior performance. However, this approach also has several advantages. First, it allows greater level of interpretability, and the development of a parsimonious theoretical model. Second, it could be more practical to implement in terms of computational infrastructure and data privacy compared to say LLM based methods. Finally, the strong performance relative to supervised approaches that have access to the ground truth suggest that even the naïve scenario described in the paper has excellent performance.
>
> Regardless of the exact methods used, our paper provides evidence for the following two insights.
>
> **Insight 1. The importance of reference information**, defined as external sources that are “out-of-distribution” compared to the documents you want to label.  A priori, it is not obvious that to label documents of a particular type, it would be useful to gather reference information from a different source, as the resulting labels can be very noisy. A more traditional approach would be to have experts label the unlabeled documents themselves. Our work shows that reference information, which are typically long documents (like Wikipedia articles or SOP’s), can often be easily gathered and provide a good starting label for the unlabeled corpus. Our work uses Multinomial Naïve Bayes (MNB) to generate these weak labels, but this can be replaced with many other techniques including other classification algorithms (e.g., Logistic Regression), nearest neighbors with embeddings like BERT, or even few-shot prompting with LLM’s if context window length permits. We chose MNB because it provides the most interpretability, which aided our mathematical modeling in Section 5.
>
> **Insight 2. The need to improve these weak labels through an iterative re-training process.** It is not obvious why such iterative re-training is necessary or useful. Again, our work chooses MNB because it provides the most interpretability: we see that as we re-train, the model learns from new features (i.e., words) that may not have appeared in reference information. In Appendix D.2, we show an example of how this iterative process corrects a document’s label fom the class sports to technology.
> This insight generalizes beyond our methodology of MNB: Imagine asking an LLM to do sentiment analysis—why would you then want to re-train a model on these LLM produced labels? Our paper argues that there is value in doing so, as it fine-tunes the boundary between positive and negative sentiment. For example, the LLM may not understand that certain tokens (e.g., slang) are extremely negative in meaning, so it misclassifies a text snippet that contains many slang words as positive. In the iterative re-training process, we see that the slang words are highly correlated with other negative words that the LLM did know, hence we learn about the negative slang words and correct those misclassified texts. Of course, if the LLM were perfect to begin with, then re-training would not be useful, but it is rarely the case that the initial weak labels will be perfect.
>
> Our paper’s claims that the DCRI framework is useful, and provides evidence through empirical study on real data, empirical study on synthetic data, and theorem for a special case. We believe that these insights would be valuable to the TLMR audience for this very relevant problem of document classification.
>
> Thank you again for reviewing our work.

---

### Review · Reviewer_ZMLa · 2025-05-21

**Summary Of Contributions:**

The paper proposes Document Classification using Reference Information (DCRI), a novel framework that leverages external reference documents to produce weak labels for document classification tasks. These labels are iteratively improved using supervised learning and optionally enhanced with limited expert feedback. The paper includes empirical evaluations on one industrial dataset (pharmaceutical deviations) and two public datasets (BBC and 20 Newsgroup), a generative model explaining the framework's behavior, and a theoretical result demonstrating the benefit of iteration in a simplified setting.

**Audience:**

Yes

**Claims And Evidence:**

Yes

**Requested Changes:**

- Include modern baselines using large language models (LLMs) or pretrained embeddings to strengthen empirical comparisons.

- Provide ablation studies on reference document length, noise, and class imbalance to further understand when DCRI works best.

- Explore alternatives to Naive Bayes (e.g., logistic regression or neural networks) within the iterative framework to demonstrate modularity more concretely.
- Offer more guidance or heuristics on how practitioners can select or curate reference documents effectively.

**Strengths And Weaknesses:**

***Strengths***
- Novel and Practical Idea: The use of reference information (e.g., Wikipedia or SOPs) to generate weak initial labels is original and potentially highly useful in domains with limited labeled data. This concept provides a new angle on semi-supervised learning by replacing initial human-labeled data with curated external documents.

- Modular and Interpretable Framework: The DCRI pipeline is modular, interpretable (via bag-of-words and Naive Bayes), and flexible enough to be adapted with more modern NLP tools, which the authors explicitly acknowledge.

- Empirical Effectiveness: The experiments are thorough, with clear comparisons to supervised, semi-supervised, and unsupervised methods. DCRI performs impressively even without labeled examples and improves further with a small number of expert labels.

- Theoretical and Simulated Support: The authors back their empirical results with a generative model (the Noisy Vocabulary Model) and a formal theorem demonstrating the benefit of the iterative update process. This adds strong credibility to the proposed approach.


***Weaknesses***
- ***Limited Scope of Baselines***: While the paper compares against traditional unsupervised, supervised, and semi-supervised methods, it lacks comparisons with more recent approaches such as self-training with pretrained embeddings (e.g., BERT-based pseudo-labeling) or prompt-based classification using LLMs.

- ***Outdated Techniques in Main Pipeline***: The use of bag-of-words and Multinomial Naive Bayes is justified for interpretability, but results would be more compelling if the authors had also demonstrated the framework's effectiveness with modern embeddings (e.g., BERT, Sentence-BERT) and classifiers (e.g., fine-tuned transformers).

- ***Dependence on Good Reference Data***: Although this is acknowledged and analyzed, the method is only effective if reasonably relevant and high-quality reference documents can be curated. This assumption may not always hold in some domains, and the method’s sensitivity to the reference set quality should be more clearly quantified.

- ***Theoretical Result Is Narrow in Scope***: The theorem applies to a highly simplified setting with only two classes and assumes infinite data. While useful for intuition, it offers limited practical guarantees in real-world settings.

---

> ### Author Response · Authors · 2025-06-03
> **Response to Reviewer**
>
> Thank you for the thorough review of our paper—we sincerely appreciate the feedback. We acknowledge that the paper could benefit from more direct discussion of state-of-the-art methods, and we also plan to expand discussion of limitations of the paper accordingly. We provide some justification below.
>
> The paper aims to provide describe an algorithmic framework that is modular in that one could use different methods in each of the respective components. Indeed, in the paper we describe a scenario with rather naïve methods such as bag of words and MNB, and it is reasonable to hypothesize that more advanced methods will lead to superior performance. However, this approach also has several advantages. First, it allows greater level of interpretability, and the development of a parsimonious theoretical model. Second, it could be more practical to implement in terms of computational infrastructure and data privacy compared to say LLM based methods. Finally, the comparison to ground truth and supervised approaches suggest that even the naïve scenario described in the paper has excellent performance.
>
> Regarding the limited scope of baselines and outdated techniques in the main pipeline, we agree that a more comprehensive study would include other methods for the modular components of DCRI, namely generating weak labels and the iterative re-training process. As you mentioned, the choice of bag of words and MNB is for interpretability as the insights from the real-world datasets motivates our theoretical model and the simulation study that followed. We believe that our insights should generalize to more sophisticated methods for generating weak labels and iterative re-training. The latter is clear, as a neural network would certainly improve over MNB on iteratively improving the labels. For the former, while it is possible in certain applications to achieve a better initial label accuracy with a more sophisticated method, our paper’s focus is on instances where only a weak label is possible. Hence, our choice of a weaker method for generating the initial labels leads to more generalizable insights across other domain applications.
>
> Regarding the reliance on good reference data, our empirical study in Section 5 show insights regarding how the quality of reference information (modeled by the proportion of vocabulary words that are corrupted) affect the quality of the final labels. We show the insight that “a little bit of quality can go a long way”, emphasizing the need for practitioners to gather high-quality reference information.
>
> Thank you again for reviewing our work!

---

### Decision · Action_Editor_KdE2 · 2025-07-09

**Recommendation:** Reject

**Additional Comments:**

The paper introduces a novel and interpretable framework for document classification in annotation-scarce settings, supported by solid theoretical justifications and promising empirical results within a constrained scope. However, the reliance on older techniques (e.g., bag-of-words, multinomial naive Bayes) and the absence of comparisons with more modern approaches (e.g., zero-shot or few-shot learning, contrastive learning, or LLM-based classifiers) limits the paper’s relevance and impact. These shortcomings, while not undermining the internal validity of the results, reduce the contribution’s alignment with current expectations in the field.

A more comprehensive evaluation including recent baselines would significantly strengthen the paper’s contribution.

**Audience:**

No

**Audience Explanation:**

All reviewers agree that the paper tackles a timely and practically important problem, but it relies on comparison with methods that are too old (BoW, multinomial naive Bayes, etc.), which severely limits the audience of the paper.

**Claims And Evidence:**

Yes

**Claims Explanation:**

The submission presents a carefully constructed framework (DCRI) for document classification in low-supervision scenarios. The methodology is theoretically motivated and empirically validated within a defined experimental scope, which relies on traditional models such as bag-of-words and multinomial naive Bayes. Reviewer ZMLa finds the claims well-supported and highlights the rigor and clarity of the evidence presented. However, both reviewers Kmjd and RS7L raise concerns that this scope is too narrow given the current state of the field.

The paper does not claim state of the art (SOTA), so I believe that the answer to Claims and Evidence should be 'Yes'.